

# 1 Modelling of water and energy exchanges over a
# 2 sparse olive orchard in semi-arid areas.

**Wafa Chebbi** [1,2], **Vincent Rivalland** [2], **Pascal Fanise** [2], **Aaron Boone** [3], **Lionel Jarlan** [2], **Hechmi**
**Chehab** [4], **Zohra Lili Chabaane** [1], **Valérie Le Dantec** [2], **Gilles Boulet** [2]
[1] Université de Carthage/INAT/LR GREEN-TEAM, 43 avenue Charles Nicole, Tunis 1082, Tunisie
[2] CESBIO, Université de Toulouse, CNES/CNRS/INRA/IRD/UPS, Toulouse, France
[3] Centre National de Recherches Météorologiques, Toulouse, France
[4] l'institut de l'olivier, Unité spécialisée de Sousse, Rue Ibn Khaldoun, Sousse 4061, Tunisie
* Correspondence: chebbiwafa@outlook.fr; Tel.: +21628003882
**Abstract:** In the Mediterranean basin, olive orchards occupy a large fraction of agricultural lands
due to its sustainability to harsh conditions, drought in particular. Since most modeling tools to
simulate vegetation functioning are not meant to represent very sparse crops (i.e., rainfed olive trees
have a vegetation fraction cover ranging from 2 to 15 %), computing the water needs and the
vulnerability to drought of an olive orchard is a challenge. There is indeed a very high contribution
of the bare soil signal to the total fluxes, and it is difficult to decipher the contribution of the tree
from that of the entire surface. In this context, in an attempt to study the olive tree hydrological
functioning at field scale (38 ha), an experimental site was setup and a Soil-Vegetation-Atmosphere
(SVAT) model has been applied. To represent the orchard soil-plant-atmosphere interactions, a
simulation with default settings was assessed using parameters derived from both the literature and
ground measurements. In this default configuration, neither the predicted actual nor the potential
transpiration could reach the observed transpiration acquired during the wet season ($R^2$=0.67, the
Root Mean Square Error (RMSE)=5.63 mm week$^{-1}$). We show that the model fails to reproduce the
relevant leaf surface that transpires. To address this issue and to improve the estimate of the year-
to-year variability of the olive tree transpiration, we propose guidance on how a SVAT model can
be modified to more appropriately represent the hydrological functioning of a sparse orchard. Once
the tree transpiration is accurately simulated ($R^2$=0.93, RMSE=1.62 mm week$^{-1}$), we evaluated
whether the fully coupled (single patch) or a fully uncoupled (two patch) system better reproduced
the total fluxes and their components. Owing to the independent characteristics of the soil columns
inherent in the assumption of the 2-patch version, the bare soil column shows a deficiency if the
topsoil root extraction is not accounted for. We deduced that we cannot accurately reproduce the
soil evaporation in this configuration. This study open perspectives for a better representation of
water fluxes over sparse tree crops into both hydrological and SVAT models.
**Keywords:** SVAT; SURFEX ISBA; MEB; olive orchard; sparse; energy and water budgets

## 35 1. Introduction

For sparse agrosystems, it is difficult to describe the exchanges at the soil-plant-atmosphere
interfaces with classical one dimensional (vertical) water and energy transfer models when large
areas of bare soil and dense vegetation islands exist side by side as is often the case in arid and semi-
arid environments (Cammalleri et al., 2013; Daamen and McNaughton, 2000; Hunt et al., 2002).
Furthermore, the wide and the deep root systems are inaccessible to classical measurement
approaches, difficult to study and poorly understood (Rossi et al., 2011), especially in very low
density and rainfed orchards when roots can extend to several meters from the stem of the tree, both
horizontally and vertically (Paltineanu et al., 2016). There is indeed a very high contribution of the
bare soil signal to the total fluxes, and it is difficult to decipher the contribution of the tree from that
of the entire surface (Hu et al., 2018; Piayda et al., 2017). Over these systems, the modeling tools are
limited by 1) the striking contrast between the bare soil and the canopy temperature response and 2)



the complex way the available energy is dissipated and partitioned between the sensible and latent
heat fluxes.
Several authors such as Cammalleri et al. (2010) suggested the application of SVAT (soil
vegetation atmospheric transfer) models to avoid the use of crop-dependent coefficients (Mata-
González et al., 2005; Subedi and Chávez, 2015), which are poorly understood for discontinuous
canopies. Evapotranspiration in rainfed agrosystems is closely controlled by the amount of rainfall
and its high spatial variability which has in turn an important impact on the convective systems.
SVAT models, which solve simultaneously the water and the energy budgets, are thus suitable tools
for exploring the plant-atmosphere exchanges. In addition to their useful application in climate
simulation in terms of the coupling with the atmosphere and the partitioning into latent and sensible
heat fluxes (Koster and Suarez, 1994), SVAT models could serve as a basis for up-scaling purposes
from plot to region (Debruyckere et al., 1997). For example, to derive evapotranspiration in catchment
hydrology, Olchev et al. (2008) used a regional process-based "SVAT-Regio" model that includes a
regionalization of meteorological information and a temporal reconstruction of the diurnal variability
of meteorological parameters for each grid cell within the study area. Selective integration of grid cell
fluxes in space and time allows estimating the energy and water fluxes for e.g. ecosystems,
catchments or entire study area for different periods from 1 day to several years. Through these
models, the availability of climatic simulations for the current century paves the way to anticipate
and to test the vulnerability of orchards to drought prediction by various future scenarios in sensitive
areas. An adequate simulation tool could also help to support farmer decision-making (e.g.
supplementary irrigation…). Traditionally, the representation of the evapotranspiration is based on
the so called "big leaf" hypothesis of the Penman-Monteith (PM) approach (Monteith, 1965), where
the land surface is treated as one homogenous layer and a single resistance is used for modeling all
sources (soil, leaves) of heat transfer. However, for partially or sparsely vegetated canopies, the PM
model may be inappropriate because the big leaf assumption is not fulfilled (i.e., the sources/sinks
for heat fluxes occur for very different conditions, esp. of temperature) (H. J. Farahani and W. C.
Bausch, 1995). These models do not provide any partition between soil evaporation and the plant
transpiration, even despite its obvious significance to the estimation of the water use over sparse
systems and implications for agricultural drought assessment. Some authors have restricted the
composition of the land covers of only two patches juxtaposed side-by-side (one unshaded bare patch
and one vegetated patch with its underlying shaded soil) and with little interaction between the two
components. In fact, the main assumption of those models is to define the sparse vegetated covers as
two disconnected sources of vegetation and soil, which are thermally uncoupled and do not exchange
water (Kustas and Norman, 1997).
By contrast, the development of two-source models (Shuttle-worth and Wallace, 1985) that
include the energy balance of the soil has improved the modeling of heat and mass exchanges for
sparsely vegetated surfaces. Baldocchi et al. (2000) recommend treating those cover types as a dual
source system. The main assumption states that the vegetation is uniformly distributed over a surface
(Raupach and Finnigan, 1988). Although this approach shows a fairly good performance for partially
covered herbaceous surfaces (low Leaf Area Index, LAI) for which turbid medium theory holds
(Raupach and Finnigan, 1988), it might no longer be valid when vegetation is clumped with dense
isolated canopies such as big trees where unshaded bare soil areas are sufficiently large to interact
directly with the atmosphere with only a limited influence of the nearby vegetation. Indeed, this
heterogeneity may change substantially the energy being absorbed by the plant canopy and the
substrate below it. In fact, a large part of the substrate will be in direct sunlight while the other part
will be in the shadow of the canopy. The resulting system might therefore exchange differently than
the same system with a uniform screening. The simulation of radiative transfer using turbid medium
assumption (Beer-Lambert law) might perform poorly since the vegetation fail to screen a large
fraction of the soil. Several studies (Anderson et al., 2005; Kustas and Norman, 1999) argue that
clumped sparse vegetation might intercept less radiation than vegetation homogenously dispersed
with the same LAI. An effective LAI is obtained by multiplying the observed LAI by a clumping
index, which describes the non-random 3D distribution of foliage. This index was incorporated by





Kustas and Norman (1999) in the thermal-based Two-Source Energy Balance (TSEB) model as the
ratio between the real canopy gap fraction and its equivalent in homogenous conditions. The beam
extinction coefficient is estimated assuming that the leaf angle distribution is ellipsoidal. The sparse-
crop model of (Shuttleworth and Wallace, 1985) has also been extended to several multilayer models
such as the Clumping (C) model (Brenner and Incoll, 1997). The C model overcomes the limitation of
uniformly distributed vegetation over a surface in the S–W model and considers three sources from
where LE is transferred to the atmosphere: the canopy, the soil under canopy, and the soil between
rows. Further than the radiation exchange, for most of these one- or two-dimensional models, the
stand evapotranspiration partitioning (i.e., the soil evaporation and the vegetation transpiration) is
obtained by weighing the whole flux by the cover fraction of each component. Usually, the cover
fraction is also deduced from the Beer-Lambert law (Brenner and Incoll, 1997; Cammalleri et al., 2010).
Some authors such as Taconet et al. (1986) distinguished two different partition factors: one for the
radiation partition (i.e., an average shielding factor) and another for momentum partition, which is
expressed as a function of LAI. In these two cases, total fluxes such as ET are thus not necessarily
equal to the already weighted sum of its components from the bare soil patch and the vegetated
patch. This simplified modelling (i.e., abstraction of the vegetated canopy as a turbid medium)
neglected the spatial separation of individual tree crowns forms. Thus, many authors proposed
to calculate the transpiration as the separate sums of sunlit and shaded fractions, weighted by their
respective leaf area within the canopy (Ding et al., 2014). A more realistic solution is provided by
three-dimensional (3D) models, which consider the canopy as an array of 3D cells characterized by
an individual LAI, which scale up water and carbon exchanges from the leaf to the canopy within
complex covers. Sinoquet et al. (2001) designed the model Radiation Absorption, Transpiration and
photosynthesis (RATP, (Sinoquet et al., 2001)) to simulate the spatial distribution of radiation and
leaf-gas exchanges within vegetation canopies as a function of canopy structure, canopy microclimate
and physical and physiological leaf properties. The spatially explicit 3D model MAESPA computes
also radiation absorption, photosynthesis and transpiration at the scale of a leaf within the crown of
individual trees within a stand, using spatial and temporal leaf-level biochemical properties linked
with stomatal gas regulation and the Penman–Monteith equation (Bowden and Bauerle, 2008).
However, 3D models are complex to implement, time and data consuming and currently cannot be
embedded in regional scale LSMs (Menenti et al., 2004).
Sparse vegetation covers (i.e., the partially covered vegetation during first stages of growth of
some crops, row crops …) are widespread but show also a great diversity in heterogeneity
levels/types. Compared to sparse herbaceous plants, sparse woody plants are developed vertically.
In particular, contrary to natural ecosystems, the trees are planted in rows in regular tillage farming
systems. Rainfed olive is a typical example of such sparse agrosystems. Under semi-arid conditions
that are typical to the Mediterranean basin, the rainfed olive trees (*Olea europaea*) are largely planted
with traditional management practices that consist in decreasing the planting density to improve the
soil volume explored by the roots (Connor et al., 2014). The lower the rainfall is, the higher the
distance between the trees will be. The required SVAT model should be adapted to the woody
vascular system of the olive and to their specific geometrics. In particular, the water use strategy of
such orchards and their adaptive properties must be taken into account.
There are two main concerns in our case: 1/ the very low fraction cover in the study site equal to
7 %, which can be regarded as bare soil, results in low fraction of net radiation available to the
vegetation if the partitioning is based on this horizontal projection fraction. It seems also that "big
leaf" potential evapotranspiration derived from most SVAT models, which use the vegetation
fraction cover as weighting factor for the turbulent fluxes partition, do not allow to achieve a
sufficient order of magnitude compared to the observed one. For example, to simulate a transpiration
value equivalent to the maximum of 3 mm day$^{-1}$ recorded during the wet period over the same study
site (Chebbi et al., 2018), a potential amount of 3/(fc=0.07)=42 mm day$^{-1}$ is required. Knowing that the
observed transpiration was accurately checked in (Chebbi et al., 2018) while compared to the
difference between the observed evapotranspiration and evaporation, there is a clear deficiency in
the modeled potential transpiration rate to represent the contribution of transpiration to the whole





area in the case of fraction cover partitioning. Clumping index would even decrease this fraction.
Moreover, the order of magnitude of our observed transpiration rate falls in the range documented
in the literature (Moreno et al., 1996; Tognetti et al., 2006). Indeed, Santos et al. (2018) reported mean
transpiration of 1.5 mm per day with maximum values observed in the summer under deficit
irrigation treatment over 10 years old olive trees with a spacing of (4.2×8 m) in southern Alentejo,
Portugal. Similarly, Moriondo et al. (2019) validated their model (dedicated to the simulation of
growth and development of olive trees) against a set of data collected over a rainfed olive grove in
Italy with ground cover of 0.19. In their research, it was also found that the simulated as well as the
observed transpirations reach 3 mm day$^{-1}$ on July. The area average transpiration is clearly stemming
from a larger surface than what can be classically computed from a turbid medium with clump LAI
of woody trees (roughly 3) weighted by the fraction cover, and must be calculated by aggregating a
larger leaf-atmosphere interacting layer. 2/ Overall, two surface schemes (coupled and uncoupled
sources) are available to represent the energy and water transfers over discontinuous canopies.
However, each of them contains some truth in our particular but widespread cover. Indeed, there is
no clear-cut recommendation to choose the appropriate scheme surface for very sparse trees. Boulet
et al. (1999) compared both surface descriptions (one compartment also named "series", and two
compartments, i.e. "patch") and pointed out that this second configuration better simulates the
energy balance for very heterogeneous covers and provides especially a more realistic estimate of
unshaded soil and vegetation individual skin temperatures and the corresponding shaded soil
temperatures. They proposed a sparseness index that helps to select the appropriate configuration.
This index is a function of the tree height and width and the spacing between rows. However, Blyth
and Harding (1995) applied both configurations (patch and series) for the tiger bush experimental
site during the HAPEX-Sahel experiment in Niger, and demonstrated that the horizontal heat transfer
between the unshaded bare soil and the vegetation is significant and is not taken into account in the
uncoupled configuration. Likewise, Aouade et al. (2019) demonstrated that the convective fluxes and
the evapotranspiration partition is also better reproduced based on a coupled approach for
moderately sparse irrigated orchards. Lhomme and Chehbouni (1999) clarified also the inconsistency
in some models describing sparse covers, in terms of convective transfers, because rough elements
(vegetation or trees) have an impact on the turbulent processes over the unshaded bare soil and vice
versa, and this interaction could not be neglected.
The overall objective of this study is to better understand, through SVAT modeling and a
detailed experimental dataset (Chebbi, 2017), the thermo-hydric functioning of the olive tree under
the present climatic conditions and above all the evolution of this functioning under predicted climate
changes. The ISBA (Interaction Sol-Biosphère-Atmosphère) model, used here, is a part of the
Externalized SURFace (SURFEX) modeling platform (Masson et al., 2013). Its first version based on a
single energy balance approach was built by Noilhan and Planton (1989) and it has been recently
extended to represent multiple energy balances in a coupled manner by Boone et al. (2017a).
Therefore, its patch parameterization is classically the same one used to represent the sub-grid
variability in land surface models coupled with atmospheric models (such as the SURFEX platform)
(Masson et al., 2013). The model is used by a large number of communities (global and regional
climate, hydrology …) and over a large range of covers. Consequently, our work contributes to the
improvement of the ISBA parameterization within the particular context of isolated trees in semi-arid
areas. The choice of this model for the present study can be justified by its ability to test future
scenarios based on future climate forcing and to predict the cover response to more recurrent drought
periods. It is also a complete physical model, that enables the comparison of the two configurations
(coupled/series and uncoupled/patch) and includes different soil water transfer schemes (i.e., force-
restore and multilayer diffusion).
The model was applied using parameters determined from observations or from the literature.
First, the different ISBA outputs were compared with observed data, including an analysis of an
inconsistency when dealing with the observed vegetation fraction cover of 7 % as the weighting of
the evaporation E and transpiration T components. Then, the model was slightly revised to address
some inconsistencies, principally the evapotranspiration partition. In particular, we look how the



effective area that transpires can be increased to match that observed T. Finally, the second issue deals
with the choice between the patch (or uncoupled) approach and the layer (or coupled) approach and
which is the configuration that better reproduce the water and energy exchanges, with respect to the
vegetation sparseness and structure of this discontinuous canopy.

## 2. Materials and Methods

### 2.1. Study site

The experimental field is a rainfed olive grove in Kairouan, central Tunisia, which is a semi-arid
area. Chebbi et al. (2018) provided a detailed description of the site. Details about the instrumentation
and the database are available online (Chebbi, 2017). The plants are spaced on a regular grid of 20 m
by 20 m (about 25 trees ha$^{-1}$ and a fraction vegetation cover of 7 %) and the mean canopy height is
maintained at about 5 m. The soil is loamy sand, with an average content of clay, silt and sand of
8 %, 4 % and 88 %, respectively.
The study was conducted during three successive hydrological years from 2013 to 2016 (starting
from September to August) at a half-hourly time step. The setup consisted of instrumented towers
with adjacent pits, a tall one close to the tree and a shorter one over the bare soil at the center of a
square delimited by four trees (including the one that is instrumented). Therefore, one tower/pit
couple is dedicated to the tree functioning and another is related to the bare soil at the inter-row. For
the meteorological data, the air temperature and the relative humidity were sampled above the tree
at 9.5 m height and above bare soil at 2.4 m height. Both wind speed and wind direction were
measured using a 3D sonic anemometer (Campbell S. CSAT3, USA) in the highest tower and a wind
transducer (Young RM 05103, USA) over the bare soil. The rainfall was collected through a rain gauge
(Campbell S. SBS-500, USA).
For the energy budget components, the vegetation net radiation was estimated using NR01 net
radiometer (Husekflux, Delft, the Netherlands) installed above the olive tree. The bare soil net
radiation was measured by an NR-Lite Net Radiometer (Kipp and Zonen, Delft, Holland). At the
orchard scale, the whole net radiation was obtained by a weighted average of vegetation and bare
soil data by their fraction covers. Similarly, the soil heat flux below the tree (i.e., in the shade for the
most part of the morning till late in the afternoon) and the soil heat within the unshaded bare soil
were determined separately using heat flux plates (Hukseflux HFP01, Delft, the Netherlands)
inserted at a depth of 2 cm. The turbulent heat fluxes were derived from the eddy covariance method
and the flux source area contains a mean vegetation fraction of about 7 %. With the aim of partitioning
the latent heat flux, the transpiration was measured continuously by inserting TDP50 sensors in the
4 trees surrounding the flux tower. The bare soil evaporation was reconstructed from the observed
soil water content at the top 5 cm (see Chebbi et al. (2018) for further details).
The surface temperatures above the vegetation and above the bare soil were derived from
thermal radiometer IR120 (Apogee Instruments Inc., Logan, UT, USA) measurements.

### 2.2. Model application and parametrisation

In order to mimic the surface heterogeneity, the ISBA model is based on the tiling method that
consists in dividing the surface area into as many homogenous entities as vegetation types juxtaposed
side-by-side in one grid. The term "patch" is used to designate this sub-grid variability, and each
patch is described by a single source approach (i.e., one single energy budget). The output fluxes are
aggregated and transferred to the atmosphere meteorological model (such as the Application of
Research to Operations at Mesoscale (AROME) model (Seity et al., 2011)).
This tile version of ISBA allows implementing easily the uncoupled (patch) configuration. One
bare soil patch represents the unshaded soil and the other one the vegetated area with the underlying
shaded soil (Fig. 1b). This configuration is based on the assumption that the turbulent mixing at the
plant-atmosphere interface occurs without disturbing the physical processes of the exposed bare soil.
The two components are thermally uncoupled and do not exchange water (Kustas and Norman,
1997). In that case, there is no radiation exchange between the exposed bare soil patch and the



vegetation patch, and each one receives the whole amount of incoming radiation and precipitation
forcing. The water and heat soil transfers are computed by solving the dynamic equations driving
the evolution of the temperature and soil water content profiles in the soil. After computing the soil
water budgets separately for each patch, total fluxes are determined through weigthing the soil and
vegetation fluxes by the relative area of each patch.
To characterize the soil and the vegetation functioning, input parameters are those classically
used in most land surface models (LSM) such as the vegetation fraction cover, the canopy height, the
minimum stomatal resistance, the Leaf Area Index, the albedo, the soil hydrodynamic parameters,
the soil layer depths, the root fraction for each layer and the aerodynamic roughness. The patch
number corresponds to one of the 19 plant functional types proposed in the model library. This
vegetation classification is in line with the Ecoclimap table (an available database for ecosystems
types that provide a consistent set of land surface parameters) which can be used for a standard
application of ISBA. The model scheme has been adapted to include soil multilayer diffusion option
(Boone et al., 2000; Decharme et al., 2011) in order to represent the heterogeneous vertical distribution
of soil properties. The soil layer number is defined according to the observed soil layer characteristics
(hydrodynamic and thermal). The soil water transfers are controlled through the retention and
hydraulic conductivity curves (Noilhan and Planton, 1989). The soil hydrodynamic parameters are
counted using the soil texture (sand and clay percentage) and 4 empirical pedo-transfer functions are
proposed while applying Brooks and Corey (1964) and Clapp and Hornberger (1978) formulations.
The added value of the multilayer scheme is the specification of root vertical distribution and the
ability to model the strong near-surface gradients of soil moisture and temperature.
For the 1P configuration (Fig. 1a) which is implemented here using the dual-source Multi-Energy
Budget version (MEB) of ISBA model developed by Boone et al. (2017) and Napoly et al. (2017), the
most relevant difference corresponds to the manner in which the vegetation and the bare soil
interactions with the atmosphere are represented. The two sources are fully coupled. The infinite thin
layer of vegetation that covers the bare soil controlled the absorbed, the reflected and the transmitted
incoming radiation through the shielding factor veg. The root extraction is then extended to the bare
soil fraction and is computed at the surface as well as at depth as a component of the soil water
balance. The distinctive features of this version are the use of the multi-layer solar radiation transfer
scheme (Carrer et al., 2013) and the resolution of multiple energy budgets at the surface of one patch,
which are coupled with each other and with the atmosphere. The radiative transfer is still weighted
by the fraction cover veg and is based on the Beer-Lambert law.
Insert Fig. 1 here
For the rest of the paper, while the coupled/series configuration is referred to as "1P", the
uncoupled/patch configuration is referred to as "2P". The "2P-BG" and "2P-VEG" designed the bare
soil patch and vegetation patch of the 2P configuration, respectively. The model inputs for the two
configurations are displayed on Table 1, in which the ground measurements are the main source of
the model inputs. The forcing data (the global and the atmospheric radiation, the humidity and the
temperature of the air, the speed and the direction of the wind, the atmospheric pressure and the
rain) are determined in-situ.
Insert Table 1 here
For the soil discretization, the number of layers and depth were defined in agreement with
the heat and water measurement depths. The vertical soil texture was prescribed for all layers
according to observations (Table 2).
Insert Table 2 here
While all the other parameters remain equal, the LAI (Leaf Area Index) is the parameter that varies
between both simulations. For the 2P simulation, we consider the LAI on the vegetated patch ($veg$=1)
which is computed as the ratio between the leaf area and the area of the soil below the tree (also
named "clump LAI"). However, for the 1P configuration, the LAI includes the area of soil which is
not covered by vegetation and is expressed as:

$$LAI = veg \times CLAI + (1-veg) \times 0 \tag{1}$$

Where the CLAI is the clump LAI and is thus equal to LAI/veg.





In addition, due to the presence of the vegetation (i.e. rough elements on a smoother substrate),
the roughness of the bare soil in the coupled configuration needs an adjustment to represent the extra
shear stress. Therefore, Raupach (1992) proposed a method for calculating the effective roughness
and displacement height of a set of scattered rough elements (made up of isolated obstacles such as
trees, shrubs, etc.). This method assimilates the rough elements to cylinders of known width (b) and
height (h) compatible to the height of vegetation and located on average at a defined distance from
each other (D). This description makes it possible to synthesize the turbulence screen in a roughness
density that characterizes the landscape. The starting point of the method is how the shear stress
changes together with the wind profile when adding one, then two and then n rough cylindrical
elements over the substrate.
**3. Results and discussions**
*3.1. The energy balance*
The modeling of the water and energy fluxes over heterogeneous covers, like our olive groves,
faces many significant issues related to both the low LAI and the complex 3D structure of the trees
(Unland et al., 1996). One of the main purposes of the study is to simulate the energy budget
components over such sparse cover. The closure of the observed energy budget was already checked
and the errors in the measurements were discussed in our previous study (Chebbi et al., 2018). Taking
into consideration the negligible LAI value, we first attempted to consider the orchard as a bare soil
in ISBA and investigated whether this assumption can provide a fairly realistic simulation of the
energy fluxes. The model was not able to track the seasonal dynamics particularly for the latent heat
flux which decrease sharply after each rain event (not shown). In addition, the RMSE between the
observed total fluxes over the orchard and the simulated fluxes from the sole bare soil patch were
significant about 31.46, 73.24, 58.23 and 44.12 W m$^{-2}$ for Rn, G, LE and H, respectively. This highlights
the interest in representing appropriately the tree which has a major impact on the fluxes. The results
of the simulation that includes the water and energy exchanges of the tree are shown in the Figure 2.
Over the whole study period, the simulated energy fluxes were plotted against the observed one in
scatterplots of the diurnal mean values of each component. The correlation coefficients of each
simulation are also provided (Fig. 2).

Insert Fig. 2 here
For all the energy budget terms, the RMSE between observations and simulations do not exceed
the range of the measurement error and the current acceptable threshold of 50 W m$^{-2}$ (Wilson et al.,
2002), with lower values for the heat soil flux and the latent (21.21 W m$^{-2}$ and 24.06 W m$^{-2}$
respectively). The sensible heat flux is the dominant turbulent flux compared to the latent heat flux
that always remains lower than 100 W m$^{-2}$.
*3.2. how representative the fraction cover is to partition evapotranspiration into evaporation and*
*transpiration?*
Once the energy balance was checked, the evapotranspiration partitioning is set as one of the
goals of our study (Fig. 3). The observations show three contrasting years: a dry year (2013), a wet
year (2014) and a moderately dry year (2015). The weekly dynamics of the evapotranspiration were
plotted and the comparison between measurements and simulations were shown (Fig. 3). In
comparison with the partitioning observations, we note that the results were less satisfactory than
the total energy budget components. Although the whole evapotranspiration was well reproduced
with an RMSE of about 4.53 mm per week and a coefficient of determination equal to 0.64, the ISBA
configuration, based on the observed set of vegetation and soil related parameters described above,
showed a poor performance and failed to quantify its two components separately: the evaporation
from the bare soil and the transpiration from the tree (Fig. 3).
Insert Fig. 3 here





The figure 3 shows that the tree transpiration is highly underestimated during the wet year,
which explains the high values of RMSE, while the two dry years show better results. In order to
check whether the maximum rate of transpiration, that is when the root zone soil moisture is high
enough to prevent water stress, the potential transpiration rate for the entire surface as computed
with the low value of vegetation fraction (0.07) was plotted (Fig. 3b). This means that, during the two
dry years 2013 and 2015, the simulated evapotranspiration corresponds numerically to the potential
rate instead of a high level of tree water stress one would expect during those dry years, and during
the wet year 2014, the simulated potential rate of evapotranspiration is far below the observed
maximum of transpiration (3 mm day$^{-1}$). The good results in 2013 and 2015 might thus be for the
wrong reason (moisture limited transpiration rate that corresponds to the potential rate of a 7 %
relative transpiration area).
The significant difference between the simulated and the observed transpiration can mainly be
related to an unrealistic proportion of transpiration area to the entire surface. The Beer-Lambert law,
which is usually used to represent a uniform layer of small reflective elements, may not work here
and fail to represent the very dense foliage located over a small area. In addition, contrarily to natural
ecosystems such as African savannah (i.e., where there is water use competition between the trees
and the grass growing after rain events), the orchard bare soil, is regularly ploughed and as a
consequence the water present in the whole unsaturated zone except for the shallow surface is
available exclusively for the olive tree.
To overcome this issue, in our case with only 7 % of vegetation fraction cover (almost a bare
soil) that represents the limit of the applicability domain of the model, we will try to adjust artificially
the appropriate parameters as an attempt to fit the observed transpiration without changing the
model formulation. For this purpose, to increase the potential and the actual transpiration, the first
assumption was to increase the effective area of leaves that transpires by testing various vegetation
fraction covers. A sensitivity study was provided in Table 3.

Insert Table 3 here

The minimum RMSE on transpiration simulations corresponds to the *veg* equal to 0.28
(equivalent to the observed *veg* multiplied by 4). This corresponds roughly to the ratio between a
transpiring area seen as a disk (projected area) in the case of 2D vegetation covers and a sphere (real
transpiring surface) for the isolated trees. This factor 4 was also reported by Lang (1991), where the
Cauchy theorem (i.e, the surface area of any convex body is equal to four times its silhouette average
area) was applied to estimate the surface area of pine needles.
Figure 4 shows the simulated actual and potential transpiration rates after increasing *veg*.
Though the transpiration results were improved at the beginning of the wet season, it is not the case
for the whole period. The predicted potential transpiration values rise properly and reach precisely
the observations during the wet period. Although the simulated transpiration fits the measured
transpiration during the first dry season 2013/2014, the model underestimates the transpiration and
simulates too much water stress during the second part of the wet season 2014/2015 and after.

Insert Fig. 4 here

*3.3. The need to represent the water supply*
Cammalleri et al. (2013) found that the Penman-Monteith model well reproduces the olive tree
functioning under moderate water conditions but is unable to reach the evapotranspiration level
during dry periods over an olive orchard in Sicily with a vegetation fraction cover of about 0.35. These
findings are also consistent with our previous results in Chebbi et al. (2018). Chebbi et al. (2018)
demonstrated that there is a lack of closure of the top first meter soil water balance of the olive grove
in the observations. More specifically the sum of the soil storage and the evapotranspiration far
exceeds the rainfall amount during dry periods. For example, the relevant rainfall events occurring
in the winter and the spring of 2014 refill the deep soil horizons and maintain high values of
transpiration even in the next summer and autumn of 2014, where there is a deficit of the soil water
balance (i.e., the sum of the soil storage and the evapotranspiration minus the rainfall is equal to 200
mm). Ramos and Santos (2009) illustrate the use of a larger amount of water for transpiration (404



mm) than the rainfall amount (240 mm) over a dry-farming olive grove as a result of an adaptation
to severe water stress conditions. In our agricultural system, this can be explained by an upslope
mound (infiltration strip) designed for water harvesting and the slope of the plot which is slight but
sufficient to promote lateral water redistribution. On top of that, a geophysical survey conducted
over the study site (not shown) proves that there is a discontinuity at 2 m depth related probably to
a less permeable layer and/or moisture accumulation. In the region of Bouhajla, a region near our
experimental site, Kanzari et al. (2012) also found that there is a semi-permeable silty-clay layer
located around 2 m which moderates the water infiltration and causes the creation of a water-
saturated layer during the wettest periods. To support this assumption, figure 4 shows that even after
increasing the vegetation fraction cover, the observed transpiration fits the potential curve during the
dry summer of 2014. This means that the tree has access to water.
This additional deep water supply was quantified on the basis of the deficit of the top meter soil
energy balance as reported by Chebbi et al. (2018). The associated value of about 200 mm was divided
arbitrarily by the number of time steps from June to August 2014. The model was slightly adjusted
by adding this amount of water onto the deeper soil layer, at each time step, as a steady source term
to the water budget.
Insert Fig. 5 here
The simulated transpiration with this water supply of 200 mm better fits the observed
transpiration (Fig. 5), as evidenced by the decrease in the RMSE to 1.62 mm week$^{-1}$ and the increase
in the determination coefficient to 0.93. This confirms our assumptions of the deep root extraction
during dry seasons. One can note also that this additional water do not prevent the plant from
suffering from water stress in the following season, which is correctly reproduced by the simulation.
*3.4. The coupled or the uncoupled scheme?*

Now that the transpiration level is correctly reproduced, the two configurations (1P and 2P)
were tested to obtain adequate physical representation of transfers from this sparse vegetation
canopy. In this context, a careful analysis of the soil evaporation, as a relevant component of the water
budget in this cover, and a comparison of the energy balance components thereafter provide us with
the possibility to select one of the two surface schemes.
For the soil evaporation, to account for the uncertainty in the bare soil functioning due to the
imperfect pedo-transfer functions, a calibration was carried out on the soil hydrodynamic properties
of each layer. The main objective of this analysis is to explore whether one can decrease the RMSE
between the simulated and observed evaporation rates due to inaccurate soil transfer parameters.
Therefore, different pedo-transfer functions were tested. Those are proposed by the model and
described above and combined with a large set of sand and clay fractions to produce a range of
corresponding values of soil conductivity and potential matric at saturation. An optimization, based
on the minimization of RMSE between the measured and the observed daily soil evaporation, was
carried out to define the percentage of sand for the different soil layers and the corresponding soil
hydrodynamic parameters (not shown). While changing the soil properties, the choice of the
percentage of sand as a key parameter for the calibration is explained by the need to maintain a
consistent global behavior of the textural classes. Although a lower RMSE was recorded for a
maximum percentage of sand (100 %) for the 2P simulation, this simulation was not consistent with
the observations and produced very high evaporation rates. Hence, no matter what the soil
hydrodynamic parameters were, the model overestimated the evaporation and the minimum
difference between the cumulative observed transpiration and that of simulated transpiration could
reach 115 mm. In addition, the RMSE was about 0.56 mm per day and the coefficient of correlation
was about 0.25 (Fig. 6).
The figure 7 illustrates an RMSE for the 1P simulation (1.66 mm week$^{-1}$) which is lower than that
of the 2P simulation (2.21 mm week$^{-1}$) (Chebbi et al., 2018). The correlation coefficient at weekly
timescale is about 0.59 and 0.46 for the 1P and the 2P simulations, respectively. Both versions of ISBA
(1P and 2P) used the same formulation of the evaporation efficiency as the method to reconstruct
evaporation in Chebbi et al. (2018), but in the later the top soil water content is taken from





observations, while in ISBA the soil moisture content is modeled. The main difference between the
simulated surface soil moisture by the 1P and the 2P versions is the consequence of the lack of roots
in the bare soil patch for the 2P simulation.
Indeed, the reality is that the olive trees roots are distributed across the entire inter-row area and
thus the whole bare soil fraction (see the root density observation in (Chebbi et al., 2018)). There is
even an increase in the root density at the mid-row spacing which is in accordance with the overlap
between the neighboring tree roots. There is also some root extraction in the top soil layer because
the soil water content measurement near the tree trunk and in the inter-row bare soil at 5 cm depth
provides similar results. Another way to confirm this finding is to estimate approximately the amount
of water extracted by the tree from the bare soil patch (i.e., the observed root extraction in the first
soil layer weighted by the fraction of transpiring area $f_c$ and divided by the bare soil relative fraction
$(1-f_c)$. So, it is not surprising that the difference between the over-simulated evaporation and this
estimated amount of water extracted by the tree is consistent with the reconstructed soil evaporation
of the bare soil patch (Fig. 6). In conclusion, it appears that splitting the cover into two different soil
water budgets is not representative of the water transfer occurring in reality.
Insert Fig. 6 here
From the energy balance point of view, the results of the two configurations that include the tree
functioning are shown in the figure 8. Regarding the simulation of the total surface energy budget
components, the series configuration provides better results than that of the patch one. Since the
albedo value was forced to the model, the small spread in the net radiation can be explained by the
surface temperature gap between the two configurations. In contrast, the RMSE of the ground heat
flux from the 2P configuration is higher than the 1P configurations and exceeds the current acceptable
threshold of 50 W m$^{-2}$ (Wilson et al., 2002). This is explained by the weighted average (by a low
vegetation fraction cover) of a soil heat flux from hot and dry unshaded bare soil, which increases the
conductive fluxes, and another from the vegetation patch where the convective fluxes are dominant.
Therefore, the resultant flux is close to the bare soil flux and less consistent with in-situ observations.
The 1P configuration improves the simulation of the ground heat flux since it limits the incoming
radiation that reaches the bare soil. For the turbulent fluxes, the lower RMSE value was obtained from
the 1P configuration for the latent heat flux (15 W m$^{-2}$) and from the 2P one for the sensible heat flux
(38 W m$^{-2}$). The improvement of the latent heat flux scores compared to the initial ones gives us more
confidence in our assumptions. In fact, it appears that splitting the cover into two patches with no
interaction at the aerodynamic level (i.e., uncoupled convective fluxes scheme) does not reflect the
real turbulent transfers.
**5. Conclusions**
A simulation of the energy and water budgets was carried out with the ISBA LSM model over
very sparse olive orchard with only 7 % of vegetation fraction cover. To evaluate the added value of
each of them in the comprehension of the ecosystem hydrological functioning, two schemes (series
and patch) already available on the model were tested. In our case, although we found that the series
configuration is more adequate to reproduce the total fluxes, it failed to partition the
evapotranspiration components. As a step towards the improvement of the model performance over
such areas, several assumptions were made and the relevant parameters were adjusted. We assumed
here that the vegetation fraction cover measured at nadir-view cannot be representative of the 3D
structure of the olive tree and the dense foliage that transpires. In the alternative, we proposed to
vary the vegetation fraction cover in order to increase the transpiration until reaching the
observations order of magnitude. The result is to consider not the evaporative surface of a disc
corresponding to the classical projection of the crown on the ground, but that of the surface of a
sphere that matches roughly the tree crown. The model, after this adjustment, gives overall
satisfactory results but tends to simulate higher water stress than the observed one during dry
seasons. This was justified by a deep accumulation of water because of the low slope of the land and
the increase of the clay fraction of a deep layer. The partitioning results were improved without
altering the consistency of the other outputs as compared to observations. The uniqueness and the



effectiveness of the olive tree transpiration process revealed by this exploratory study calls for further measurement and modeling studies in this unusual and interesting environment.

As in the research of Kennedy et al. (2019), which implements plant hydraulics in the Community Land Model (CLM5), our future study will focus on modifying the ISBA model to represent some processes typical to large woody species such as an improved stomatal functionning more adapted to the semi-arid context, better stress functions, the variations in xylem water storage, and more importantly the root system and its hydraulic redistribution through sap flow sensors in line with previous works of (Nadezhdina et al., 2015).

**Acknowledgments:**

Financial support from the MISTRALS/SICMED program, the ALTOS European project of PRIMA program, the ERANET-MED CHAAMS project, the FLUXMED European project of WATER JPI program and the CNES/TOSCA program for the PITEAS project are gratefully acknowledged. This research is carried out in the frame of the NAILA international laboratory.

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

**Figures**



(a)                                                              (b)

Fig. 1: the 1P (a) and 2P (b) configurations






Fig. 2: scatterplots between the observed and the simulated energy budget terms during the study period ((a): Rn; (b): G; (c): LE and
(d): H)




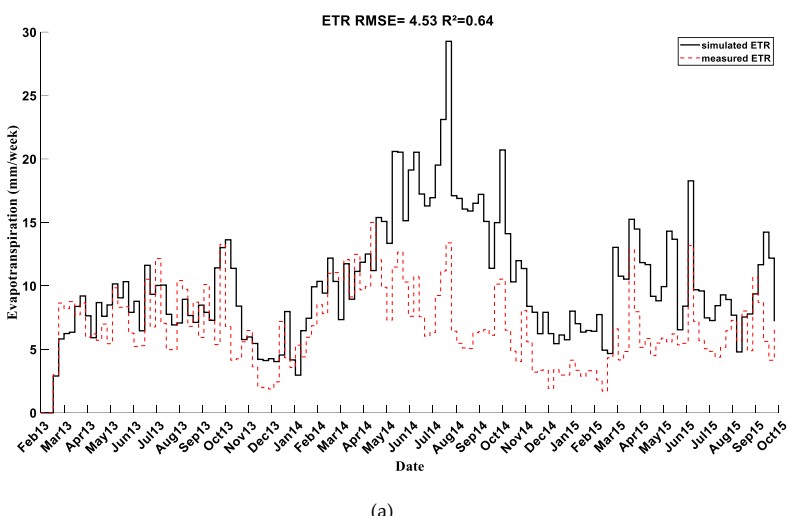

(a)

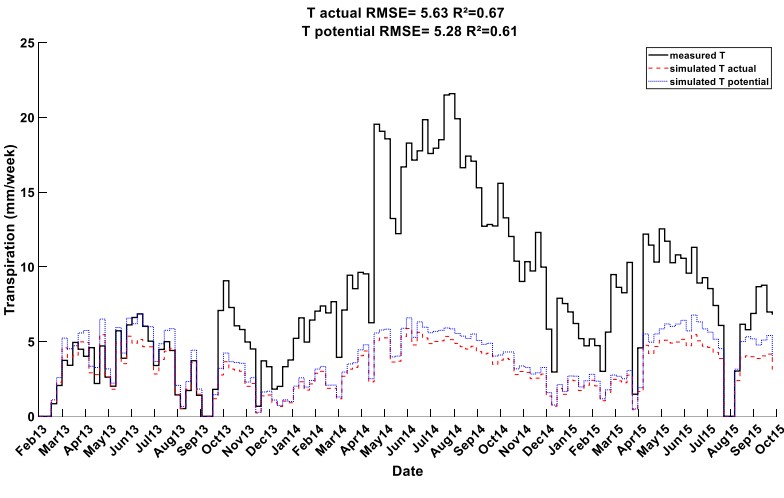

(b)





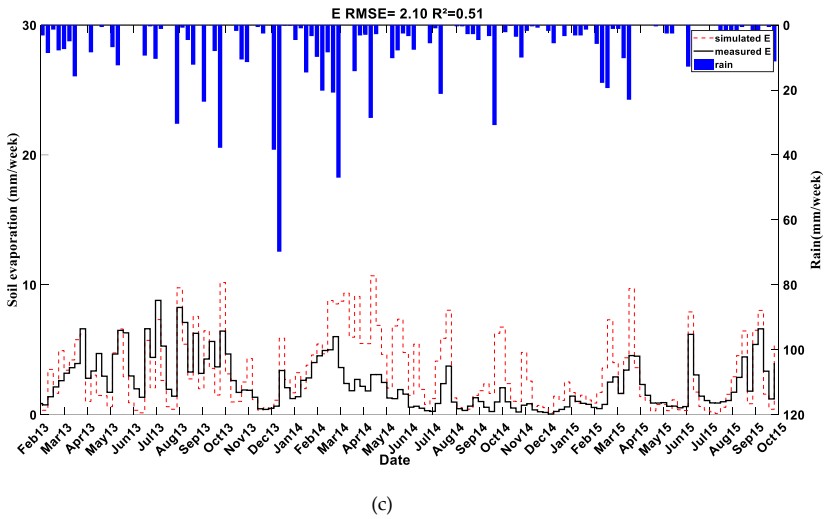

(c)

Fig. 3: the weekly evapotranspiration and its partitioning and their associated scores; (a) ETR: total evapotranspiration, (b) T: the actual and the potential transpiration and (c) E: bare ground evaporation


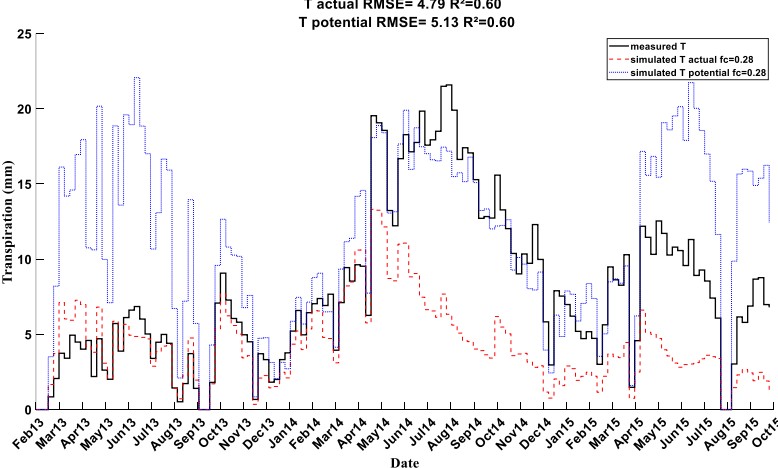

Fig. 4: a comparison between the actual and the potential transpiration with a fraction cover $f_c$ of 0.28






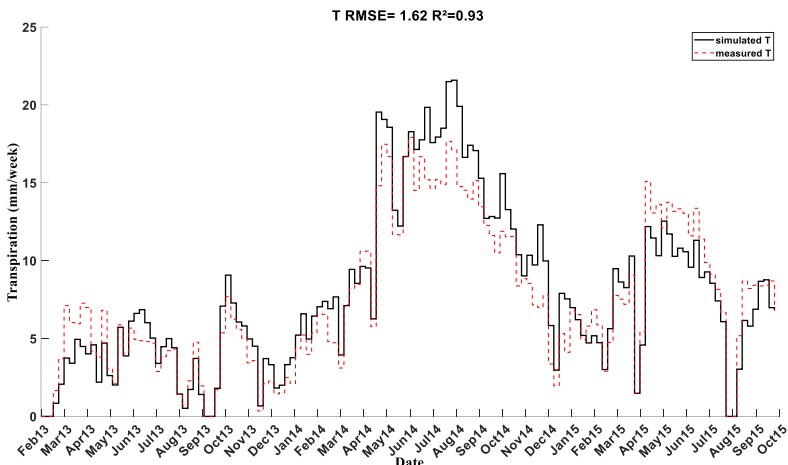

Fig. 5: a comparison between the measured and the simulated transpiration after the water supply at a weekly scale and their related statistical scores


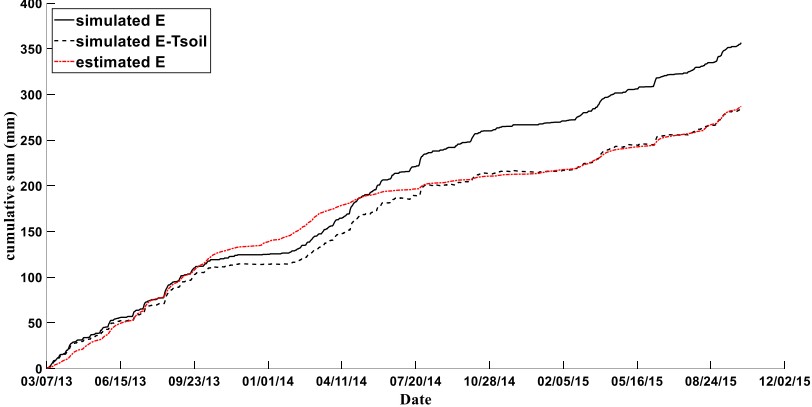

Fig. 6: the cumulative sum of the simulated evaporation from the 2P run , the simulated evaporation minus the estimated water extracted by roots from the bare soil top layer (Tsoil) and the determined evaporation from soil water content measurements




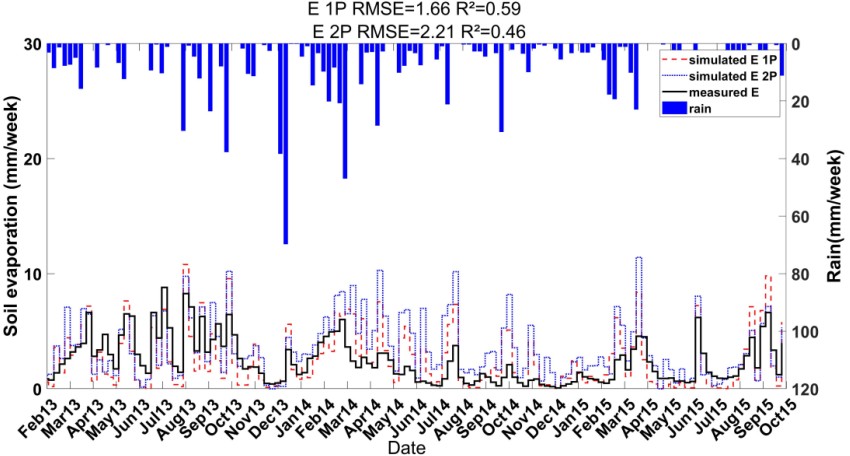

Fig. 7: the simulated evaporation from the 1P and the 2P run after increasing the vegetation fraction cover and adding the water supply.


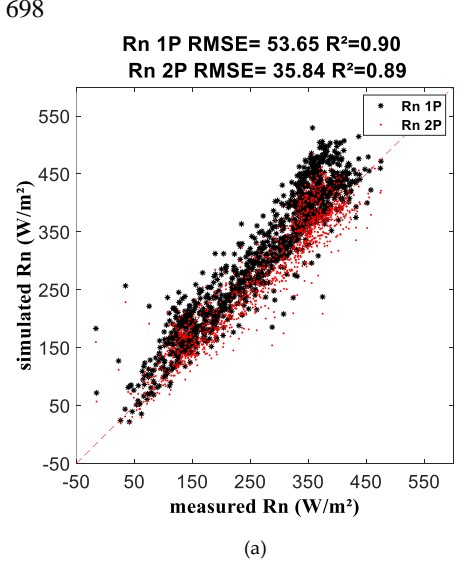

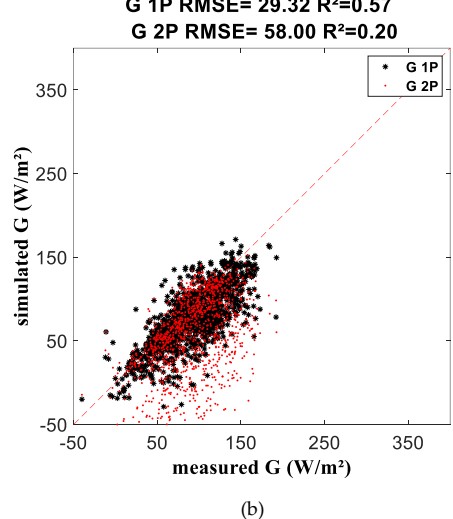

(a)
(b)





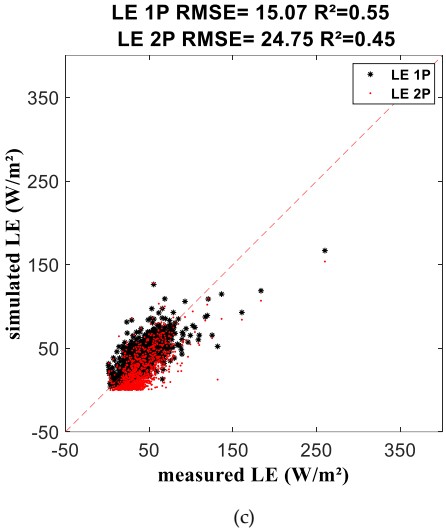

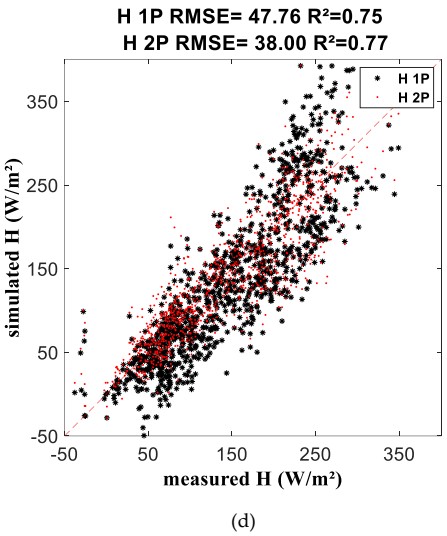

Fig. 8: scatterplots between the observed and the simulated energy budget terms ((a): Rn; (b): G; (c): LE and (d): H) derived from the 1P and the 2P configurations

















# Tables

Table 1: The initial SVAT input parameters

| Parameter | Value | Source |
|---|---|---|
| Layers number | 6 | Observations |
| Layer depth | 0.075;0.225;0.4;0.75;1;2 m | - |
| Soil depth | 1 m | Observations |
| water content at saturation | 0.35 | Observations |
| Field capacity | 0.15 | Observations |
| Wilting point | 0.05 | Observations |
| Soil albedo | 0.32 | Observations |
| Vegetation albedo | 0.28 | Observations |
| Soil emissivity | 0.96 | Literature (Rubio et al., 1997) |
| Vegetation emissivity | 0.98 | Literature (Rubio et al., 1997) /default value |
| $Z_0$ | 0.52 m for 1P<br>0.02 for 2P-BG<br>0.73 for 2P-VEG | Estimated according to (Raupach, 1992) modified by (Verhoef, 1995)<br><br>Literature (Garratt, 1994) |
| $Z_{om}/Z_{oh}$ | 50 | Default value |
| *veg* | 0.07 | Observations |
| LAI | 3.2 m²/m² of soil | Observations |
| Tree height | 5.67 m | Observations |
| Minimum stomatal resistance | 160 m/s | Literature (Dbara et al., 2016) /default value |
| Root fraction | 9;20;18;13;8;40 % | Observations to a depth of 0.6 m and extrapolation to 2 m depth based on literature (Moreno et al., 1996) |

Table 2: the measured soil texture between 0 and 1m depth

| depth (m) | Sand (%) | Clay (%) | Silt (%) |
|---|---|---|---|
| 0 | 88.2 | 8 | 3.8 |
| 0.1 | 88.2 | 8 | 3.8 |
| 0.2 | 88.2 | 8.5 | 3.3 |
| 0.4 | 81.1 | 14.5 | 4.4 |
| 0.6 | 79.9 | 13.9 | 6.2 |
| 0.8 | 80.7 | 13.6 | 5.7 |
| 1 | 88.2 | 7.3 | 4.5 |
| 1.6 | 88.2 | 7.3 | 4.5 |

Table 3: the scores between the simulated and the measured transpiration for different vegetation fraction covers

| *veg* | RMSE | R² | NASH |
|---|---|---|---|
| 0.07 | 1.24 | 0.53 | -43.55 |





| | | | |
|---|---|---|---|
| **0.14** | 1.03 | 0.43 | -9.62 |
| **0.21** | 0.99 | 0.32 | -5.37 |
| **0.28** | **0.89** | **0.21** | **-4.35** |
| **0.35** | 0.94 | 0.11 | -3.83 |
| **0.42** | 0.99 | 0.05 | -3.58 |
| **0.49** | 1.03 | 0.02 | -3.46 |







# Annex 1: the model description


For the surface energy budget, the net radiation ($R_n$), the sensible heat flux (H) and the latent
heat flux (LE) are expressed in W/m² as follows:

$$R_n = R_g(1 - \alpha) + \varepsilon(R_{atm} - \sigma T_s^{\,4}) \tag{1}$$

$$H = \rho_a C_p C_H V_a(T_s - T_a) \tag{2}$$

$$LE = L_v(E_g + E_v) \tag{3}$$

$$E_g = (1 - veg)\rho_a C_H V_a[h_u q_{sat}(T_s) - q_a] \tag{4}$$

$$E_v = veg\,\rho_a C_H V_a h_v[q_{sat}(T_s) - q_a] \tag{5}$$

Where $R_g$ and $R_{atm}$ are the global and the atmospheric radiation respectively, $\alpha$ is the combined
soil/vegetation albedo, $\varepsilon$ is the total surface emissivity weighted by the vegetation cover (*veg*), $\sigma$ is the
Stefan-Boltzmann constant, $T_s$ is the total surface temperature, $\rho_a$ is the air density, $C_p$ is the air specific
heat, $V_a$ is the wind speed, $T_a$ is the air temperature, $C_H$ is the drag coefficient, $L_v$ is the latent heat of
vaporization, $q_{sat}(T_s)$ is the saturated specific humidity at the temperature $T_s$, qa is the atmospheric
specific humidity, $h_u$ is the evaporation efficiency which depends on the top soil layer water content,
$h_v$ is the Halstead coefficient which is meant to represent both the leaf intercepted water evaporation
and the plant transpiration $h_v = (1 - \delta)R_a/(R_a + R_s) + \delta$, $R_a$ is the aerodynamic resistance and $R_s$ is
the surface resistance.
The surface resistance that monitors the transpiration is defined by (Jarvis, 1976) and controlled by
the minimal stomatal resistance parameter $R_{smin}$.
The surface heat flux (G) corresponds to the residual term of the energy budget equation and is
expressed as follows:

$$G = R_n - H - LE \tag{6}$$

The surface temperature, which is associated with the soil temperature at the top soil layer, depends
on *veg*, the surface heat flux and the heat characteristics of the layer below. The thermal gradients at
the surface and within the soil are governed by the Fourrier law and are written as follows:

$$\frac{dT_s}{dt} = C_T[G - \frac{\bar{\lambda}_1}{\Delta z_1}(T_s - T_2)] \tag{7}$$

$$\frac{dT_i}{dt} = \frac{1}{C_{g_i}}\frac{1}{\Delta z'_i}[\frac{\bar{\lambda}_{i-1}}{\Delta \bar{z}_{i-1}}(T_{i-1} - T_i) - \frac{\bar{\lambda}_i}{\Delta \bar{z}_i}(T_i - T_{i+1})]; \qquad \text{i=2…n} \tag{8}$$

Where $\Delta z'_i = (\Delta z_i + \Delta z_{i+1})/2$ is the layer i depth, $\Delta \bar{z}_i$ is the spacing between the nodes of the layers i
and i-1, $C_{g_i}$ is the layer-averaged soil heat capacity, $\bar{\lambda}_i$ is the inverse-weighted arithmetic mean of the
soil thermal conductivity at the interface between two consecutive nodes.
For the description of the soil water transfer, the model is based on the Richards equation on its
mixed form using both state variables: the soil water content and water pressure head. This equation
is applicable independently of the saturation state in addition to its privilege in the modeling of
heterogeneous soil property (texture) profile.
By analogy with the thermal gradient resolution, the liquid-vapor exchanges of soil water are
written as follows:

$$\frac{dw_1}{dt} = \frac{1}{\Delta z_1}[-\bar{k}_1(\frac{\psi_1 - \psi_2}{\Delta \bar{z}_1} + 1) - \bar{\vartheta}_1\left(\frac{\psi_1 - \psi_2}{\Delta \bar{z}_1}\right) + \frac{S_1}{\rho_w}] \tag{9}$$

$$\frac{dw_i}{dt} = \frac{1}{\Delta z_i}[\left(F_{i-1} - F_i + \frac{S_i}{\rho_w}\right) \qquad \text{i=2…n} \tag{10}$$





$$F_i = \bar{k}_\mathrm{i}\left(\frac{\psi_i - \psi_{i+1}}{\Delta \bar{z}_i} + 1\right) + \bar{\vartheta}_\mathrm{i}\left(\frac{\psi_i - \psi_{i+1}}{\Delta \bar{z}_i}\right) \tag{11}$$


Where $S_i$ is the soil-water source (infiltration) /sink (soil evaporation and root extraction) term,
$\psi_i$ is the soil matric potential, $\bar{k}_\mathrm{i}$ is the geometric mean of soil hydraulic conductivity, $\bar{\vartheta}_\mathrm{i}$ the
geometric means of the isothermal vapor conductivity.
(Boone et al., 2017b) developed the dual-source Multi-Energy Budget version (MEB). The
distinctive features of this version are the use of the multi-layer solar radiation transfer scheme and
the resolution of multiple energy budgets at the surface of one patch, which are coupled with each
other and with the atmosphere. The energy budgets at the surface are expressed as prognostic
equations governing the dynamics of the bulk vegetation canopy $T_v$, for ice and snow free conditions.

$$C_v \frac{\partial T_v}{\partial t} = R_{nv} - H_v - LE_v \tag{12}$$

$$C_{g,1} \frac{\partial T_{g,1}}{\partial t} = R_{ng} - H_g - LE_g - G_{g,1} \tag{13}$$

Where $T_{g,1}$ is the uppermost surface soil temperature, $L_f$ is the latent heat of fusion (J.kg⁻¹). g
refers to the ground, v to the vegetation and c to the interface between the canopy air space and the
vegetation.
For the water budget for the uppermost soil layer, the equation is:

$$\rho_w \Delta z_{g,1} \frac{\partial w_{g,1}}{\partial t} = P_r - P_{rv} - D_{rv} - E_g - R_0 - F_{g,1} \tag{14}$$

Where $w_{g,1}$ is the uppermost soil water content layer, $P_r - P_{rv}$ is the remaining rainfall after
interception, F is the soil water vertical flux, $E_g$ is the ground evaporation, $R_0$ is the surface runoff and
$D_{rv}$ is the canopy drip of liquid water.
The different fluxes are expressed as a function of resistances ($R_a = \frac{1}{V_a C_H}$) in s.m⁻¹ instead of the
dimensionless heat and mass exchange coefficient ($C_H$). The resistances represent the water extraction
efficiency at the soil-plant-atmosphere interfaces.
The sensible heat fluxes are defined as follows:

$$H_{v} = \rho_a \frac{(T_v - T_c)}{R_{av-c}} \tag{15}$$

$$H_{g} = \rho_a \frac{(T_g - T_c)}{R_{ag-c}} \tag{16}$$

$$H_c = \rho_a \frac{(T_c - T_a)}{R_{ac-c}} \tag{17}$$

Where $\rho_a$ is the lowest atmospheric layer average air density, $T_c$ is the specific temperature of
the canopy air space.
Though the sensible heat fluxes (i.e. the H variables) that are expressed in terms of temperature
herein for simplicity, thermodynamic variables such as potential temperature or dry static energy are
used in the actual model computations (see Boone et al. (2017))
776       Similarly, the three water vapor fluxes are determined as:

$$E_{v} = \rho_a h_{sv} \frac{(q_{satv} - q_c)}{R_{av-c}} \tag{18}$$

$$E_{g} = \rho_a \frac{(q_g - q_c)}{R_{ag-c}} \tag{19}$$





$$E_{c=}\rho_a \frac{(q_c - q_a)}{R_{ac-a}}$$

(20)

Where $q_c$ is the specific humidity of the canopy air space and $h_{sv}$ is the Halstead coefficient for
the canopy evapotranspiration.
The radiative transfer is based on the Beer-Lambert law and the heat conduction fluxes are
defined in ISBA-MEB and for ISBA referring to Eq. 6, Eq. 7 and Eq. 8. In ISBA, Ts represents the mixed
surface temperature (soil and vegetation) and the thermal inertia coefficient ($C_T$) is used. In MEB,
despite $T_{g,1}$ equivalence to $T_s$, it only have a conformity with the temperature of the bare soil. We also
tend to choose the effective heat capacity ($C_g$=1/$C_T$) in this version.
Both of the heat capacities C and the thermal conductivities $\lambda$ are functions of the organic
content and the texture of the soil.