# Peer review of "Modelling of water and energy exchanges over a sparse olive orchard in semi-arid areas."

_Hydrology and Earth System Sciences, 2020_

## Referee Comment (RC1) · Anonymous Referee #1 · 4 May 2020

This paper from Chebbi et al. deals with the modeling of fluxes exchange in a semi-arid olive orchard. The literature on the topic is quite rich, but the research topic is still relevant due to the lack of a "good-for-all" solution available in the literature at the moment. Specifically, the paper seems to aim at exploring the use of two different modeling framework, namely the single- and the two- path approaches. My main concern with the paper is the lack of clarity in the logical reasoning behind the evaluation protocol adopted here. Firstly the author evaluate the models in term of transpiration, but it is not clear which scheme they tested. Single? Two? Does it matter at this point? The authors never clarify, hence it is not possible for a reader to understand if their conclusions are reasonable. Then they come up with an empirical calibration (again, not clear related to which scheme), without accounting for possible other sources of error, and

only after all of that they discuss a comparison of the two scheme, where it is not clear how the previous calibration and correction play a role. Maybe this approach is logical if the reasoning behind is clarified in the text, but at the present the flow of the paper results really odd and difficult to follow and comment. I suggest to strongly revisit the text to make more clear to the readers why the comparison procedure was structure in this way (maybe with the support of a flow chart). At the present state, it is difficult to me to give a fair evaluation of the results without the needed context.

Major comments Introduction The introduction is too long in my opinion, and it needed to be streamlined. For instance, between lines 140 and 160 too many concept are cramped, with the result to be confusing and also to mix-up different concepts that are not meant to be used in the same modeling framework (e.g., clumping factor and fc-based partitioning). L198-206. Such details are not needed here, since a reader not familiar with the model cannot understand the content of this paragraph at this point of the text. Methods The authors should focus on the key features relevant for this study and that distinguish the 1P from the 2P. This brief, rather generic, description of the model is not useful for a reader not familiar with the model. For instance, in the 2P approach the concept of clumping factor is not relevant (since the vegetation fully cover is patch, and likely assumed to be spherically random), so the reference in the introduction to clumping factor is confusing when discussing the partitioning. The concept is relevant for the 1P, but it is not clear if accounted or not. The value of LAI reported in Table 1 (3.2) refers to the projection of the tree crown (e.g., m2 of leafs over m2 of soil covered by the projection of the tree) and it is only used in P2 (I assume), whereas the "field scale" LAI is a much different value discussed successively in Eq. (1). This is a rather key point, that is not well explained in the text. In Eq. (1) you call LAI the LAI used in 1P, as a function of the LAI used in 2P (which was previously called LAI as well), and then defining CLAI as LAI/veg. A reader may then read LAI = 3 and CLAI = 46, which is not the case, I guess. Additionally, in this discussion is never mentioned if a clumping factor is used and how it is defined. Since it is often mentioned in the introduction, it would be important to clearly state. The minimum stomatal resistance

is another key parameter, much discussed in the literature on olive trees. Many people can argue that errors in this parameters are much more likely than in the interpretation of LAI. Again, this need a lot of justification to be completed ignored here and in the discussion. L248. Please use here the term 2P and stick to 1P vs. 2P for the rest of the paper (as you stated later on, L285 but failed to apply in some circumstances). The continuous interchanged usage of path/sources make difficult for the readers to follow the rest of the text (especially because the 1P is a two-source and the 2P is parallel single sources). Results L320-325. Is this discussion really necessary? At the best, I would frame this part as a benchmark bottom minimum in term of model performance. L327. From here on it start the confusion, since no clarification on which version of the model is discussed in these figures. It is not possible to me to have a full analysis of these results if no context is given. Minor comments L11. I would suggest to replace sustainability with resistance. L13. Even if generally low, cover fraction reaches values definitely higher than that, without discussing intensive olive orchards. L15. I would suggest to replace decipher with separate/extract. L72-73. Please correct the reference format. L183. The reference here to climate change is, in my opinion, out of place. L193. This reasoning adopted to justify the use of ISBA (not needed in my opinion), is week, since almost all the models can "test future scenarios based on future climate forcing". Again, I would stick to the actual goal of the presented research, without involving climate change, which is not the focus of this study. Fig. 1. I would suggest to invert the two panels, since 1b is referred to before 1a. Table 1. Please clarify that you are talking about SOIL layers here.

---

## Author Comment (AC1) · 25 May 2020

**Authors Response to Reviewer 1 comments**

This paper from Chebbi et al. deals with the modeling of fluxes exchange in a semi-arid olive orchard. The literature on the topic is quite rich, but the research topic is still relevant due to the lack of a "good-for-all" solution available in the literature at the moment. Specifically, the paper seems to aim at exploring the use of two different modeling framework, namely the single- and the two- path approaches.

1. My main concern with the paper is the lack of clarity in the logical reasoning behind the evaluation protocol adopted here. Firstly, the author evaluates the models in term of transpiration, but it is not clear which scheme they tested. Single? Two? Does it matter at this point? The authors never clarify; hence it is not possible for a reader to understand if their conclusions are reasonable. Then they come up with an empirical calibration (again, not clear related to which scheme), without accounting for possible other sources of error, and only after all of that they discuss a comparison of the two schemes, where it is not clear how the previous calibration and correction play a role. Maybe this approach is logical if the reasoning behind is clarified in the text, but at the present the flow of the paper results really odd and difficult to follow and comment. I suggest to strongly revisit the text to make clearer to the readers why the comparison procedure was structure in this way (maybe with the support of a flow chart). At the present state, it is difficult to me to give a fair evaluation of the results without the needed context.

    *Response:*

    *We agree with the reviewer that both aspects (lack of proper representation of transpiration and 1P versus 2P representation) are treated one after the other but are mentioned all through it. We would separate both issues more clearly. When dealing with the model evaluation in terms of transpiration, all results correspond to the 1P scheme but are also valid for the 2P one, we would stress this fact. This 1P default configuration with default setting and using parameters derived from ground measurement was thus used in the first part as the reference to tackle model outputs inconsistency. Then, the model was slightly revised to address these issues for both versions (the increase of the fraction cover and the water supply assumptions) before intercomparing them. Once the evapotranspiration partition is correctly reproduced over the entire system, the comparison between the two versions of the surface scheme (1P vs 2P) was evaluated.*
    *In the revised version of the manuscript, the paragraph below would be added: "in this study, the model was run using parameters determined from observations or from the literature. First, the different ISBA outputs from 1P configuration, applied as a benchmark, were compared with observed data, including an analysis of an inconsistency when dealing with the observed vegetation fraction cover of 7 % as the weighting of the evaporation E and transpiration T components. These findings hold also true for the 2P configuration. Then, the model was slightly revised to address those inconsistencies, principally the evapotranspiration partition. In particular, we looked at how the effective area that transpires can be increased to match the observed T. an additional water supply was added to match the observed transpiration after significant rainfall amounts. Finally, the second issue deals with the choice between the patch (or uncoupled) approach and the layer (or coupled) approach and which is the configuration that better reproduce the water and energy exchanges, with respect to the vegetation sparseness and structure of this discontinuous canopy. These two configurations could finally be assessed and compared when the limitations (i.e., effective fraction cover and the water supply) arising from the first issue were cleared".*

2. The introduction is too long in my opinion, and it needed to be streamlined. For instance, between lines 140 and 160 too many concept are cramped, with the result to be confusing and also to mix-up different concepts that are not meant to be used in the same modeling framework (e.g., clumping factor and fcbased partitioning).

*Response:*

*In a revised manuscript, this paragraph would be condensed accordingly " There are two main concerns in our case: 1/ the very low fraction cover in the study site equal to 7 %, which can be regarded as bare soil, results in low fraction of net radiation available to the vegetation if the partitioning is based on this horizontal projection fraction. It seems also that "big leaf" potential evapotranspiration derived from most SVAT models, which use the vegetation fraction cover as weighting factor for the turbulent fluxes partition, do not allow to achieve a sufficient order of magnitude compared to the observed one. For example, to simulate a transpiration value equivalent to the maximum of 3 mm per day recorded during the wet period over the same study site (Chebbi et al., 2018), a potential amount of 3/(fc=0.07)=42 mm day-1 would be required. The observed transpiration was checked in Chebbi et al. (2018) through comparison to the difference between the observed evapotranspiration and evaporation. Moreover, the order of magnitude of our observed transpiration rate falls in the range documented in the literature (Moreno et al., 1996; Tognetti et al., 2006). Indeed, Santos et al. (2018)reported mean transpiration of 1.5 mm per day with maximum values observed in the summer under deficit irrigation treatment over 10 years old olive trees with a spacing of (4.2×8 m) in southern Alentejo, Portugal. Similarly, Moriondo et al. (2019) validated their model (dedicated to the simulation of growth and development of olive trees) against a set of data collected over a rainfed olive grove in Italy with ground cover of 0.19. In their research, it was also found that the simulated as well as the observed transpirations reach 3 mm per day in July. Therefore, there is a clear deficiency in the modeled potential transpiration rate to represent the contribution of transpiration to the whole area in the case of fraction cover partitioning. The area average transpiration is clearly stemming from a larger contributing surface than what can be classically computed from a turbid medium with clump LAI of woody trees (roughly 3) weighted by the fraction cover, and must be calculated by aggregating a larger leaf-atmosphere interacting layer."*

3. L198-206. Such details are not needed here, since a reader not familiar with the model cannot understand the content of this paragraph at this point of the text.

*Response:*

*The following paragraph would be removed in a revised manuscript. "The model was applied using parameters determined from observations or from the literature. First, the different ISBA outputs were compared with observed data, including an analysis of an inconsistency when dealing with the observed vegetation fraction cover of 7 % as the weighting of the evaporation E and transpiration T components. Then, the model was slightly revised to address some inconsistencies, principally the evapotranspiration partition. In particular, we look at how the effective area that transpires can be increased to match the observed T. Finally, the second issue deals with the choice between the patch (or uncoupled) approach and the layer (or coupled) approach and which is the configuration that better reproduce the water and energy exchanges, with respect to the vegetation sparseness and structure of this discontinuous canopy."*

4. Methods The authors should focus on the key features relevant for this study and that distinguish the 1P from the 2P. This brief, rather generic, description of the model is not useful for a reader not familiar with the model. For instance, in the 2P approach the concept of clumping factor is not relevant (since the vegetation fully cover is patch, and likely assumed to be spherically random), so the reference in the introduction to clumping factor is confusing when discussing the partitioning. The concept is relevant for the 1P, but it is not clear if accounted or not.

*Response:*

*The clumping factor is used here only as the mean to relate LAI for the vegetation patch in the 2P configuration (CLAI, which corresponds to the field estimated LAI of the tree crown) and the 1P one (area average LAI). This would be clarified in the revision. The description of the model would be condensed as suggested in the revision and details about model description would be moved to the Annex1.*
*The paragraph L297-302 would be modified accordingly:*
*"While all the other parameters remain equal, the LAI (Leaf Area Index) is the parameter that varies between both simulations. Indeed, for the 1P configuration, the field scale LAI (=0.24 m²/m²) includes the area of soil which is not covered by vegetation and is expressed as:*

$$LAI = veg \times CLAI + (1-veg) \times 0 \qquad (1)$$

*Where the CLAI, used for the 2P configuration, is the clump LAI (i.e., the ratio between the leaf area and the area of the soil below the tree) and is thus equal to LAI/veg (=3 m²/m²)."*

5. The value of LAI reported in Table 1 (3.2) refers to the projection of the tree crown (e.g., m2 of leafs over m2 of soil covered by the projection of the tree) and it is only used in P2 (I assume), whereas the "field scale" LAI is a much different value discussed successively in Eq. (1). This is a rather key point, that is not well explained in the text. In Eq. (1) you call LAI the LAI used in 1P, as a function of the LAI used in 2P (which was previously called LAI as well), and then defining CLAI as LAI/veg. A reader may then read LAI = 3 and CLAI= 46, which is not the case, I guess. Additionally, in this discussion is never mentioned if a clumping factor is used and how it is defined. Since it is often mentioned in the introduction, it would be important to clearly state.

*Response:*

*See response 4.*
*The LAI values would be clarified in Table 1 as follow:*

| CLAI | 3 m²/m² of soil for 1P | Observations |
|---|---|---|

*The clumping factor included in TSEB model was mentioned in the introduction but not used in our study.*

5. The minimum stomatal resistance is another key parameter, much discussed in the literature on olive trees. Many people can argue that errors in this parameter are much more likely than in the interpretation of LAI. Again, this need a lot of justification to be completed ignored here and in the discussion.

*Response:*

*The justification of the minimum stomatal resistance value would be further justified from the following literature review.*

| rs (s/m) | | age | site | article |
|---|---|---|---|---|
| | 124.4 | 18 year old | cordoba spain | (Moriana et al., 2002) |
| range from | 423.0 | 13 year old | seville spain | (Rodriguez-Dominguez et al., 2019) |
| to | 141.0 | 13 year old | seville spain | (Rodriguez-Dominguez et al., 2019) |
| | 169.2 | 8 year old | seville spain | (Hernandez-Santana et al., 2016) |
| | 166.7 | 26 year old | seville spain | (Fernández et al., 1997) |
| | 162.7 | 40-year-old | spain | (Torres-Ruiz et al., 2011) |
| | 211.5 | 7 year old | southern italy | (Giorio et al., 1999) |
| | 235.0 | 30 ans | south east of Tunis | (Dbara et al., 2016) |
| | 282.0 | 10 ans | central italy | (Marino et al., 2014) |

*The default value of Rsmin proposed by the model falls within the range of the mean rsmin value found in the literature. The lower limit of rsmin is about 124 m/s as found in the study of Moriana et al. (2002).*

*The figure below shows the potential transpiration deriving from the reference 1P configuration for two values of rsmin (the minimum value reported in the literature to our knowledge for olive trees 124 m s$^{-1}$ and the value used for this study 160 m s$^{-1}$.*

[Figure]

6. L248. Please use here the term 2P and stick to 1P vs. 2P for the rest of the paper (as you stated later on, L285 but failed to apply in some circumstances). The continuous interchanged usage of path/sources make difficult for the readers to follow the rest of the text (especially because the 1P is a two-source and the 2P is parallel single sources).

*Response:*

*Would be edited as suggested. The terminology (1P/2P) would be thus used all along the manuscript to enhance clarity and readability.*

7. Results L320-325. Is this discussion really necessary? At the best, I would frame this part as a benchmark bottom minimum in term of model performance.

   *Response:*

   *The following paragraph would be removed in the revised manuscript "Taking into consideration the negligible LAI value, we first evaluate the model in a bare soil condition as a lower limit in terms of model performance and investigated whether this assumption can provide a fairly realistic simulation of the energy fluxes. The model was not able to track the seasonal dynamics particularly for the latent heat flux which shows peaks after rainfall event followed by sharp decrease when the soil is dry (not shown). This temporal pattern of LE was inconsistent with field observations. In addition, the RMSE between the observed total fluxes over the orchard and the simulated fluxes from the sole bare soil patch were significant about 31.46, 73.24, 58.23 and 44.12 W m⁻² for Rn, G, LE and H, respectively."*

8. L327. From here on it start the confusion, since no clarification on which version of the model is discussed in these figures. It is not possible to me to have a full analysis of these results if no context is given.

   *Response:*

   *In the revised version of the manuscript, the following paragraph would be added accordingly: "The simulation, here, is a 1P configuration used as reference to evaluate the model performance and the energy fluxes deriving from this simulation are shown in the figure 2."*
   *we would also clarify in figures captions which simulation was used.*

9. L11. I would suggest to replace sustainability with resistance.

   *Response:*

   *As suggested, sustainability would be replaced with resistance.*

10. L13. Even if generally low, cover fraction reaches values definitely higher than that, without discussing intensive olive orchards.

    *Response:*

    *The sentence would be corrected as follow: "(i.e., rainfed olive trees that have a vegetation fraction cover ranging from 2 to 15 %)"*

11. L15. I would suggest to replace decipher with separate/extract.

    *Response:*

    *Decipher would be replaced with separate in the revision.*

12. L72-73. Please correct the reference format.

    *Responses:*

    *The reference format would be corrected.*

13. L183. The reference here to climate change is, in my opinion, out of place.

*Response:*

*"and above all the evolution of this functioning under predicted climate changes" this part of the sentence would be removed.*

14. L193. This reasoning adopted to justify the use of ISBA (not needed in my opinion), is week, since almost all the models can "test future scenarios based on future climate forcing". Again, I would stick to the actual goal of the presented research, without involving climate change, which is not the focus of this study.

*Response:*

*This paragraph would be modified as follow: "The choice of this complete physical model for the present study can be justified by its ability to test the two configurations (coupled and uncoupled) and the different soil water transfer schemes (i.e., force-restore and multilayer diffusion) within the same modelling environment."*

15. Fig. 1. I would suggest to invert the two panels, since 1b is referred to before 1a.

*Response:*

*The two panels would be inverted.*

16. Table 1. Please clarify that you are talking about SOIL layers here.

*Response:*

*Would be edited as suggested.*

**References**

Chebbi, W., Boulet, G., Le Dantec, V., Lili Chabaane, Z., Fanise, P., Mougenot, B., Ayari, H., 2018. Analysis of evapotranspiration components of a rainfed olive orchard during three contrasting years in a semi-arid climate. Agric. For. Meteorol. 256–257, 159–178. https://doi.org/10.1016/J.AGRFORMET.2018.02.020

Dbara, S., Haworth, M., Emiliani, G., Ben Mimoun, M., Gómez-Cadenas, A., Centritto, M., 2016. Partial Root-Zone Drying of Olive (Olea europaea var. 'Chetoui') Induces Reduced Yield under Field Conditions. PLoS One 11, e0157089. https://doi.org/10.1371/journal.pone.0157089

Fernández, J.E., Moreno, F., Girón, I.F., Blázquez, O.M., 1997. Stomatal control of water use in olive tree leaves. Plant Soil 190, 179–192. https://doi.org/10.1023/A:1004293026973

Giorio, P., Sorrentino, G., D'Andria, R., 1999. Stomatal behaviour, leaf water status and photosynthetic response in field-grown olive trees under water deficit. Environ. Exp. Bot. 42, 95–104. https://doi.org/10.1016/S0098-8472(99)00023-4

Hernandez-Santana, V., Rodriguez-Dominguez, C.M., Fernández, J.E., Diaz-Espejo, A., 2016. Role of leaf hydraulic conductance in the regulation of stomatal conductance in almond and olive in response to water stress. Tree Physiol. 36, 725–735. https://doi.org/10.1093/TREEPHYS/TPV146

Marino, G., Pallozzi, E., Cocozza, C., Tognetti, R., Giovannelli, A., Cantini, C., Centritto, M., 2014. Assessing gas exchange, sap flow and water relations using tree canopy spectral reflectance indices in irrigated and rainfed Olea europaea L. Environ. Exp. Bot. 99, 43–52. https://doi.org/10.1016/j.envexpbot.2013.10.008

Moreno, F., Fernandez, J.E., Clothier, B.E., Green, S.R., 1996. Transpiration and root water uptake by olive trees. Plant Soil 184, 85–96. https://doi.org/10.1007/BF00029277

Moriana, A., Villalobos, F.J., Fereres, E., 2002. Stomatal and photosynthetic responses of olive (Olea europaea L.) leaves to water deficits. Plant, Cell Environ. 25, 395–405. https://doi.org/10.1046/j.0016-8025.2001.00822.x

Moriondo, M., Leolini, L., Brilli, L., Dibari, C., Tognetti, R., Giovannelli, A., Rapi, B., Battista, P., Caruso, G., Gucci, R., Argenti, G., Raschi, A., Centritto, M., Cantini, C., Bindi, M., 2019. A simple model simulating development and growth of an olive grove. Eur. J. Agron. 105, 129–145. https://doi.org/10.1016/j.eja.2019.02.002

Rodriguez-Dominguez, C.M., Hernandez-Santana, V., Buckley, T.N., Fernández, J.E., Diaz-Espejo, A., 2019. Sensitivity of olive leaf turgor to air vapour pressure deficit correlates with diurnal maximum stomatal conductance. Agric. For. Meteorol. 272–273, 156–165. https://doi.org/10.1016/j.agrformet.2019.04.006

Santos, F., Santos, L., F., 2018. Olive Water Use, Crop Coefficient, Yield, and Water Productivity under Two Deficit Irrigation Strategies. Agronomy 8, 89. https://doi.org/10.3390/agronomy8060089

Tognetti, R., d'Andria, R., Lavini, A., Morelli, G., 2006. The effect of deficit irrigation on crop yield and vegetative development of Olea europaea L. (cvs. Frantoio and Leccino). Eur. J. Agron. 25, 356–364. https://doi.org/10.1016/j.eja.2006.07.003

Torres-Ruiz, J.M., Fernández, J.E., Diaz-Espejo, A., Martín-Palomo, M.J., Morales-Sillero, A., Muriel, J.L., Romero, R., 2011. Stomatal Control and Hydraulic Conductivity in "Manzanilla" Olive Trees under Different Water Regimes. Acta Hortic. 888, 149–155.

---

## Referee Comment (RC2) · Anonymous Referee #2 · 16 Jun 2020

This manuscript presents an extensive field and modeling study to test the model scheme that better describes the energy fluxes over a sparse olive grove (7% vegetation cover). I find some critical issues with the measurements and the analysis, which I think the authors need to carefully consider before this paper can be published. They are commented on the PDF.

My main concern is that the authors have attempted to test the two versions of the model on daily and weekly scales, by modifying some of the parameters. These attempts seem to be arbitrary (for example, they changed the soil texture parameters although they have measured this quantity and they know what it is). Instead, they could have looked at finer temporal resolution (hourly or sub-hourly) to unravel the processes and the dynamics that contribute to the total daily fluxes. They also showed

an agreement of ~50 W m-2 for the daily fluxes claiming these are satisfactory error magnitudes, while these magnitudes are considered satisfactory at hourly scale but not at daily scale.

Please also note the supplement to this comment:
https://www.hydrol-earth-syst-sci-discuss.net/hess-2020-104/hess-2020-104-RC2-supplement.pdf
* * *
[Figure]

**Supplement:**

[revised manuscript text omitted]

---

## Author Comment (AC2) · 14 Jul 2020

**Authors Response to Reviewer 2 comments**

This manuscript presents an extensive field and modeling study to test the model scheme that better describes the energy fluxes over a sparse olive grove (7% vegetation cover). I find some critical issues with the measurements and the analysis, which I think the authors need to carefully consider before this paper can be published.

1. My main concern is that the authors have attempted to test the two versions of the model on daily and weekly scales, by modifying some of the parameters. These attempts seem to be arbitrary (for example, they changed the soil texture parameters although they have measured this quantity and they know what it is). Instead, they could have looked at finer temporal resolution (hourly or sub-hourly) to unravel the processes and the dynamics that contribute to the total daily fluxes. They also showed an agreement of _50 W m-2 for the daily fluxes claiming these are satisfactory error magnitudes, while these magnitudes are considered satisfactory at hourly scale but not at daily scale.

*Response:*

*We thank the reviewer for his in depth reading of the manuscript.*

*As a first attempt, we choose the weekly and daily scales to ensure that daily and seasonal water and energy balances are consistent. The purpose was to test future scenarios based on projected climate forcing until 2050 in order to study the evolution of this system functioning under predicted climate changes. We will now focus on diurnal fluctuations of the energy budget components as well as its impact on the root-tree-atmosphere transfers at hourly time scales in order to better document the stomatal functioning and the hydraulic redistribution. To do so, sap flow systems are going to be installed in roots. These finer scales could also be useful in the view of the remote sensing data assimilation, particularly surface temperature.*

*The soil texture parameters were modified from measured values as an attempt to explain whether one can decrease the RMSE between the simulated and observed evaporation rates of the bare soil patch due to inaccurate soil transfer parameters (ISBA pedotransfer functions used to derive the water retention and conductivity curves), while retaining the consistency between the parameter values with respect with soil textural properties (e.g. ensuring that one keeps a sandy behavior or a more silty behavior for all soil parameters, not each one separately). But this optimization fails to decrease the sharp peak of modeled soil evaporation to the observed one after rain event even with a fraction of sand of 100 %. So, we deduced that it was not a soil moisture availability issue but a potential transpiration rate issue due to the way transpiration is computed and scaled to the whole area. Finally, the observed parameters of the soil are kept for the rest of the study. This would be mentioned in the revision manuscript.*

*For the RMSE, it should be remembered that, over this specific cover of very sparse olive orchard in semi-arid area, it would be difficult to find better results. Indeed, the fluxes shown in the Figure 8 corresponds to mean values between 9h and 16h and this would be clearly stated in the text and in the figure captions. They were considered as satisfactory because their RMSE, for example for the latent heat flux (15.07 W/m² for 1P and 24.75 W/m² for 2P) fall within the range between the hourly (50 W/m²) and the daily (20W/m²) accepted thresholds including night data as cited by Kalma et al. (2008). In fact, hourly RMSE for LE was about 47.54 W/m² and 55.27 W/m² for the 1P and the 2P configurations, respectively. Daily RMSE for LE was about 11.08 W/m² and 14.31 W/m² for the 1P and the 2P configurations, respectively. Such details would be made more explicit in the revised manuscript*

2. L67 and L130: « …) »

*Response:*

*"…)" would be replaced by ", etc)".*

3. L72: (H. J. Farahani and W. C. Bausch, 1995)

*Response:*

*The reference format would be corrected.*

4. L104: S–W not defined

*Response:*

*It would be defined as follow in the revision "the Shuttleworth-Wallace model (S–W model)"*

5. L105: LE not defined

*Response:*

*It would be defined as follow in the revision "latent heat flux (LE)"*

5. L148: through? the word "while" is misused here.

*Response:*

*As suggested, while would be replaced with "through" in the revised manuscript*

6. L140: "1/ the very low fraction cover in the study site equal to 7 %, which can be regarded as bare soil, results in low fraction of net radiation available to the vegetation if the partitioning is based on this horizontal projection fraction. It seems also that "big leaf" potential evapotranspiration derived from most SVAT models, which use the vegetation fraction cover as weighting factor for the turbulent fluxes partition, do not allow to achieve a sufficient order of magnitude compared to the observed one. For instance, to simulate a transpiration value equivalent to the maximum of 3 mm day per day recorded during the wet period over the same study site (Chebbi et al., 2018), a potential amount of (3/(fc=0.07))=42 mm per day would be required. The observed transpiration was checked in Chebbi et al. (2018) through comparison to the difference between the observed evapotranspiration and evaporation. Moreover, the order of magnitude of our observed transpiration rate falls within the range documented in the literature (Moreno et al., 1996; Tognetti et al., 2006). Indeed, Santos et al. (2018) reported mean transpiration of 1.5 mm per day with maximum values observed in the summer under deficit irrigation treatment over 10 years old olive trees with a spacing of (4.2×8 m) in southern Alentejo, Portugal. Similarly, Moriondo et al. (2019) validated their model (dedicated to the simulation of growth and development of olive trees) against a set of data collected over a rainfed olive grove in Italy with ground cover of 0.19. In their research, it was also found that the simulated as well as the observed transpirations reach 3 mm per day on July. Therefore, there is a clear deficiency in the modeled potential transpiration rate to represent the contribution of transpiration to the whole area in the case of fraction cover partitioning. The area average transpiration is clearly stemming from a larger contributing surface than what can be classically computed from a turbid medium with clump LAI of woody trees (roughly 3) weighted by the fraction cover, and must be calculated by aggregating a larger leaf-atmosphere interacting layer." This belong in the discussion rather than in the introduction. There seems to be a lack of information here. What do the models calculate? How far are they?

*Response:*

*Our choice to keep this paragraph in the introduction can be justified by the main concerns*

*that we expect to encounter in such environment with very low fraction cover and for which we try to provide an answer. Here, we explain how all SVAT models that use the vegetation fraction cover to partition evaporation and transpiration, could not in all cases reach the observed transpiration of 3 mm per day. In the paper, we want to first document and explain this mismatch by changing the input parameters in a new PFT while keeping the model structure as it is. It thus belongs to Introduction. In the Result section, the answer to the question "how far are they" is further documented. Here, we only stress as a justification of the research that no model based on fc can have a transpiration rate of 42 mm/day which would lead to the observed 3 mm per day if partitioning is on the basis of 7% cover.*

7. L198: "The model was applied using parameters determined from observations or from the literature. First, the different ISBA outputs were compared with observed data, including an analysis of an inconsistency when dealing with the observed vegetation fraction cover of 7 % as the weighting of the evaporation E and transpiration T components. Then, the model was slightly revised to address some inconsistencies, principally the evapotranspiration partition. In particular, we look at how the effective area that transpires can be increased to match the observed T. Finally, the second issue deals with the choice between the patch (or uncoupled) approach and the layer (or coupled) approach and which is the configuration that better reproduce the water and energy exchanges, with respect to the vegetation sparseness and structure of this discontinuous canopy." This is a mix of objectives and M&M. It is confusing and difficult to follow.

   *Response:*

   *As suggested by the reviewer 1, this paragraph would be removed in the revised manuscript.*

8. L210: "(Chebbi et al., 2018) provided a detailed description of the site. Details about the instrumentation and the database are available online (Chebbi, 2017)." This is not enough. I would want to know more about the site without having to go and read elsewhere. Please provide all the information you think is relevant here.

   *Response:*

   *In the revised version of the manuscript, the following paragraph would be added accordingly: "The experimental field is a rainfed olive grove in Kairouan, central Tunisia (35°18'17.14"; 9°54'56.62"). It is a semi-arid area characterized by the high spatio-temporal variability of precipitation and air temperature. The trees are 80 years old, spaced on a regular grid of 20 m by 20 m (about 25 trees ha-1 and a fraction vegetation cover of 7 %) and the mean canopy height is maintained at about 5 m. The soil is loamy sand, with an average content of clay, silt and sand of 8 %, 4 % and 88 %, respectively and a bulk density of 1.65 g cm-3."*

9. L216: « The setup consisted of instrumented towers with adjacent pits » Instrumented with what instrumentation?

   *Response:*

   *This paragraph would be modified as follow: "The setup consisted of two towers with adjacent pits, a tall one close to the tree instrumented with eddy covariance sensors and a shorter one over the bare soil at the center of a square delimited by four trees (including the one that is instrumented), both with net radiation sensors and soil heat flux plates at the surface. Therefore, one tower/pit couple is dedicated to the tree functioning and another is related to the bare soil at the inter-row.".*

10. L220: "air temperature and relative humidity were sampled above the tree at 9.5 m height and above bare soil at 2.4 m height." At 2.5 m I presume that the conditions measured were strongly influenced by the surrounding trees and did not represent the bare soil entirely.

*Response:*

*The model input data are derived from climate forcing sampled at 9.5 m. This would clearly state in the revision.*

11. L226: « the vegetation net radiation was estimated using NR01 net radiometer (Husekflux, Delft, the Netherlands) installed above the olive tree » At what height? This means you were assuming that Rn is evenly distributed across the soil and that measurement at one position is representative. I presume this is far from the reality and I wonder if you at least evaluated what might be the error introduced by this assumption.

*Response:*

*This sentence would be modified in the revised manuscript as follow: "the vegetation net radiation was estimated using NR01 net radiometer (Husekflux, Delft, the Netherlands) installed at 1.5 m above the olive tree, therefore seeing a mixed surface including canopy (80% of the Field Of View( FOV)) and bare soil (20% of the FOV)".*

12. L228: « At the orchard scale, the whole net radiation was obtained by a weighted average of vegetation and bare soil data by their fraction covers. Similarly, the soil heat flux below the tree (i.e., in the shade for the most part of the morning till late in the afternoon) and the soil heat within the unshaded bare soil were determined separately using heat flux plates (Hukseflux HFP01, Delft, the Netherlands) inserted at a depth of 2 cm» One sample for each location? Given the heterogeneity nature of soils this is problematic. Its problematic for completely bare soils, and more so when there is sparse vegetation and the shading patterns change constantly. 1. It is reported and widely accepted in the literature that inserting a soil heat flux so close to the surface is problematic. 2. How did you account for the heat stored at the uppermost 2 cm above the plate? The amount of stored energy at this layer, especially under bare soil conditions may account for 10s of percent of the total flux, and ignoring it results in a large underestimation.

*Responses:*

*Heat flux plates (Hukseflux HFP01, Delft, the Netherlands) are placed at a depth of 2 cm: 2 plates below the tree and 4 plates below bare soil. For the bare soil and the soil under the tree, 3 plates are also inserted at depths roughly equal to 5, 15 and 30 cm. Bare soil is regularly ploughed and therefore fairly homogeneous at the surface.*
*The surface soil heat flux is calculated by correcting the soil heat flux at the 2 cm depth for the 0–2 cm heat storage as:*
$$G = G_z + [C \, \Delta T / \Delta t] * \Delta z$$
*where C is the heat capacity per unit volume of the soil, and ΔT/Δt is the change of the mean temperature of the layer per unit time.*
*By doing so for the 4 heat plates at the bare soil site and the 2 plates below the canopy, one computes the soil heat flux in unshaded (Gs) and (Gv) conditions, respectively.*
*In the revised version of the manuscript, the following paragraph would be modified accordingly:*
*"Similarly, the soil heat flux below the tree (i.e., 2 plates below the tree in the shade for the most part of the morning till late in the afternoon) and the soil heat within the unshaded bare soil (4 plates below bare soil) were determined separately using heat flux plates (Hukseflux HFP01, Delft, the Netherlands) inserted at a depth of 2 cm after correcting for the 0-2cm heat storage. Bare soil is regularly ploughed and therefore fairly homogeneous at the surface."*

13. L232: «The turbulent heat fluxes were derived from the eddy covariance method». Did you apply all the relevant corrections to the flux computation? How dod you obtain the fluxes? What software did you use? What sampling frequency did you use?

*Response:*

*Data from the EC system (20 Hz sampled measurements of wind speed and gas concentration)
were collected. Quality control is analyzed using the "Eddypro" software (open source software
application developed, maintain and supported by LI-COR Biosciences). We use the software's
default settings for time-series checks, low and high pass filtering, density fluctuations, sonic
anemometer tilt correction with double rotation.
The REddyProc tool ("Biogeochemical Integration | Services / REddyProcWebGapFilling," 2011)
is used to fill gaps of meteorological and processed eddy covariance data. This method consists
in filling water fluxes gaps required to estimate monthly balances using data with similar
meteorological conditions defined via the global radiation (Rg), the air temperature (Ta) and the
vapor pressure deficit (VPD).
Some of these details would be added in the revised manuscript.*

14. L273: "Reverse the order of the figure so that the text follows it  - fig. 1a should be described
    first and then fig. 1b.".

    *Response:*

    *The two panels would be inverted.*

15. L295: «The vertical soil texture was prescribed for all layers according to observations»
    Number of repetitions?

    *Response:*

    *Three repetitions per layer.*

16. L315: «The modelling of the water and energy fluxes over heterogeneous covers, like our olive
    groves, faces many significant issues related to both the low LAI and the complex 3D structure
    of the trees (Unland et al., 1996). One of the main purposes of the study is to simulate the energy
    budget components over such sparse cover » This better suit in the introduction

    *Response:*

    *Would be moved as suggested to the introduction.*

17. L318: «The closure of the observed energy budget was already checked and the errors in the
    measurements were discussed in our previous study (Chebbi et al., 2018) » and the findings
    were that? Is it a good closure? a bad one? Again I need to go look for yet another paper to read.
    Please provide the main findings here if they are relevant.

    *Response:*

    *The closure of the energy balance for the whole three-year study is first assessed using half
    hourly values of the energy balance components for each day. The slope is about 0.91, the
    intercept is about 12.06 W m⁻² and the coefficient of determination is 0.75. As shown by*
    Anderson and Wang (2014) *and* Leuning et al. (2012)*, the energy balance closure is improved
    when aggregated to daily totals. The slope is then about 0.98 the intercept is 16.6 W m⁻² and
    the coefficient of determination is 0.83.*
    *Such details would be made more explicit in the revised manuscript.*

18. L322: «The model was not able to track the seasonal dynamics particularly for the latent heat
    flux which decrease sharply after each rain event (not shown). In addition, the RMSE between
    the observed total fluxes over the orchard and the simulated fluxes from the sole bare soil patch

were significant about 31.46, 73.24, 58.23 and 44.12 W m⁻² for Rn, G, LE and H, respectively. This highlights the interest in representing appropriately the tree which has a major impact on the fluxes » I suggest excluding this paragraph. It is obvious and does not contribute to the story.

*Response:*

*This paragraph would be removed as suggested.*

19. L333: «For all the energy budget terms, the RMSE between observations and simulations do not exceed the range of the measurement error and the current acceptable threshold of 50 W m⁻² (Wilson et al., 2002), with lower values for the heat soil flux and the latent heat flux (21.21 W m⁻² and 24.06 W m⁻² respectively). » This acceptable threshold is for hourly data not for daily averages. For daily averages these errors are rather large. They are lower as the fluxes are smaller. In percentage the LE errors are ~50%! They are very large. This is also shown in the very low r2=0.32. Did you verify that this correlation is significant?

*Response:*

*See answer to 1. Those are mean values between 9h and 16h and these fluxes correspond the run before applying the proposed adjustment for LE. This statement could be rephrased in the revised manuscript since this threshold correspond to the Latent heat flux at hourly scale.*

20. L466: «In conclusion, it appears that splitting the cover into two different soil water budgets is not representative of the water transfer occurring in reality» This is expected and somewhat trivial. In order to really get down to understanding the governing factors (and not just modify somewhat arbitrarily the model parameters, like changing the soil structure (wasn't this measured in the field? what sense does it make to change this in the model?) one probably needs to look at finer temporal resolutions. The effects and the dynamics are at half-hourly or hourly scales and not daily (or weekly).

*Response:*

*We agree with the reviewer that splitting the soil into two compartments is expected. However, here, we try to quantify the error propagation due to this cover dispatching and to show how far is it from reality since the 2P configuration is used for sparse orchard functioning description by the scientific community. In fact, the use of the patch configuration requires the introduction of a term source/sink at the root system level ensuring the compulsory exchange between the two columns.*
*See response to 1 for looking to intraday scale.*

21. L471: «the 1P configuration provides better results than that of the 2P one » I see inconsistency with RMSE increase for 2 of the fluxes and decrease for the other 2, so how can you state this? L483: « In fact, it appears that splitting the cover into two patches with no interaction at the aerodynamic level (i.e., uncoupled convective fluxes scheme) does not reflect the real turbulent transfers. » How can you justify this conclusion when only 2 of the fluxes performed better?

*Response:*

*We already justify that for the latent heat flux the coupling between the two columns is necessary for a good estimate of the total evapotranspiration and its components. However, for the net radiation, better scores were obtained from the patch configuration.*
*The scores of the soil heat flux were unexpected. In fact, G derived from the 2P configuration, with a larger fraction of sunlight bare soil than the one in the 1P run, would normally represents better the observations. Now, we could not provide additional explanations.*
*These conclusions would be modified in the revision as follow:*

*"the 1P configuration provides better results than that of the 2P one for LE"*

*"In fact, it appears that splitting the cover into two patches with no interaction does not reflect the real turbulent transfers at least for the latent heat flux. However, the 2P configuration seems to be more realistic for the net radiation. This suggests that a so-called "mixed" model (standard 1P for H and LE and soil, but patch formulation for Rn) would probably better represent the functioning of this sparse orchard."*

**References**

Anderson, R.G., Wang, D., 2014. Energy budget closure observed in paired Eddy Covariance towers with increased and continuous daily turbulence. Agric. For. Meteorol. 184, 204–209. https://doi.org/10.1016/J.AGRFORMET.2013.09.012

Biogeochemical Integration | Services / REddyProcWebGapFilling [WWW Document], 2011. URL https://www.bgc-jena.mpg.de/bgi/index.php/Services/REddyProcWebGapFilling (accessed 1.7.17).

Chebbi, W., 2017. nasrallah flux database [WWW Document]. SEDOO OMP website. https://doi.org/DOI: 10.6096/MISTRALS-SICMED.1479

Chebbi, W., Boulet, G., Le Dantec, V., Lili Chabaane, Z., Fanise, P., Mougenot, B., Ayari, H., 2018. Analysis of evapotranspiration components of a rainfed olive orchard during three contrasting years in a semi-arid climate. Agric. For. Meteorol. 256–257, 159–178. https://doi.org/10.1016/J.AGRFORMET.2018.02.020

Kalma, J.D., McVicar, T.R., McCabe, M.F., 2008. Estimating Land Surface Evaporation: A Review of Methods Using Remotely Sensed Surface Temperature Data. Surv. Geophys. 29, 421–469. https://doi.org/10.1007/s10712-008-9037-z

Leuning, R., Gorsel, E. van, Massman, W.J., Issac, P.R., 2012. Reflections on the surface energy imbalance problem. Agric. For. Meteorol. 156, 65–74. https://doi.org/10.1016/J.AGRFORMET.2011.12.002

Moreno, F., Fernandez, J.E., Clothier, B.E., Green, S.R., 1996. Transpiration and root water uptake by olive trees. Plant Soil 184, 85–96. https://doi.org/10.1007/BF00029277

Moriondo, M., Leolini, L., Brilli, L., Dibari, C., Tognetti, R., Giovannelli, A., Rapi, B., Battista, P., Caruso, G., Gucci, R., Argenti, G., Raschi, A., Centritto, M., Cantini, C., Bindi, M., 2019. A simple model simulating development and growth of an olive grove. Eur. J. Agron. 105, 129–145. https://doi.org/10.1016/j.eja.2019.02.002

Santos, F., Santos, L., F., 2018. Olive Water Use, Crop Coefficient, Yield, and Water Productivity under Two Deficit Irrigation Strategies. Agronomy 8, 89. https://doi.org/10.3390/agronomy8060089

Tognetti, R., d'Andria, R., Lavini, A., Morelli, G., 2006. The effect of deficit irrigation on crop yield and vegetative development of Olea europaea L. (cvs. Frantoio and Leccino). Eur. J. Agron. 25, 356–364. https://doi.org/10.1016/j.eja.2006.07.003

Wilson, K., Goldstein, A., Falge, E., Aubinet, M., Baldocchi, D., Berbigier, P., Bernhofer, C., Ceulemans, R., Dolman, H., Field, C., Grelle, A., Ibrom, A., Law, B.., Kowalski, A., Meyers, T., Moncrieff, J., Monson, R., Oechel, W., Tenhunen, J., Valentini, R., Verma, S., 2002. Energy balance closure at FLUXNET sites. Agric. For. Meteorol. 113, 223–243. https://doi.org/10.1016/S0168-1923(02)00109-0